# Analysis of the measurement uncertainty for a 3D wind lidar

**Wolf Knöller[1], Gholamhossein Bagheri[2], Philipp von Olshausen[1], and Michael Wilczek[2,3]**

[1]Fraunhofer Institute for Physical Measurement Techniques IPM, Georges-Köhler-Allee 301, 79110 Freiburg, Germany
[2]Max Planck Institute for Dynamics and Self-Organization, Am Faßberg 17, 37077 Göttingen, Germany
[3]Theoretical Physics I, University of Bayreuth, Universitätsstraße 30, 95447 Bayreuth, Germany

**Correspondence:** Philipp von Olshausen (philipp.olshausen@ipm.fraunhofer.de)

**Abstract.** High-resolution three-dimensional (3D) wind velocity measurements are of major importance for the characterization of atmospheric turbulence. The use of a multi-beam wind lidar focusing on a measurement volume from different directions is a promising approach for obtaining such wind data. This paper provides a detailed study of the propagation of measurement uncertainty of a three-beam wind lidar designed for mounting on airborne platforms with geometrical constraints that lead to increased measurement uncertainties of the wind components transverse to the main axis of the system. The uncertainty analysis is based on synthetic wind data generated by an Ornstein–Uhlenbeck process as well as on experimental wind data from airborne and ground-based 3D ultrasonic anemometers. For typical atmospheric conditions, we show that the measurement uncertainty of the transverse components can be reduced by about 30 %–50 % by applying an appropriate post-processing algorithm. Optimized post-processing parameters can be determined in an actual experiment by characterizing measured data in terms of variance and correlation time of wind fluctuations, allowing for the optimized design of a multi-beam wind lidar with strong geometrical limitations.

## 1 Introduction

In the atmospheric sciences, our knowledge of the atmospheric boundary layer (ABL) is mainly based on observations of turbulent flow (Garratt, 1994). Atmospheric turbulence is a complex phenomenon, with scales involved ranging from sub-meter to kilometer (Wyngaard, 2010). For large spatial and temporal scales, the ABL plays an important role in fields such as numerical weather prediction (Bauer et al., 2015), climate science (Davy, 2018), and air pollution meteorology (Quan et al., 2014). However, the focus of interest has recently shifted to smaller scales, which include microphysical aspects of clouds that are not yet sufficiently understood (Bodenschatz et al., 2010). Progress in this field is needed to further reduce uncertainties in weather models and climate projections (Bony et al., 2015; Stevens et al., 2021). To shed light on this part of the ABL there is a strong demand for highly resolved, local, and small-scale three-dimensional (3D) wind data.

Highly resolved 3D wind data can be acquired by conventional sensors such as 3D ultrasonic anemometers and multi-hole Pitot tubes. These are not remote measurement techniques, since the measurement volume is in the close vicinity of the instrument, and depending on the mounting platform the wind turbulence can be disturbed in a way that precludes measuring highly resolved wind in the ABL. Coherent Doppler lidar (light detection and ranging) is the measuring technique of choice for the remote measurement of wind, widely used for wind industry applications (Pena et al., 2013; Kumer et al., 2016; Hill, 2018; Fuertes et al., 2014; Lundquist et al., 2015). The measuring technique can be based on continuous-wave or pulsed lasers, mostly operating at 1550 nm. To resolve 3D information rather than line-of-sight information, conical scans are widely used. Such systems average over large lateral spatial and temporal scales and usually assume a homogeneous wind flow within the measuring volume (Bingöl et al., 2009), typically covering a range of tens of meters (Schlipf et al., 2020; Wilhelm et al., 2021). This precludes the measurement of complex and small-scale turbulence (Sathe et al., 2015).

A novel 3D wind lidar, the CloudKite Turbulence LiDAR (CTL), is developed by the Fraunhofer Institute for Physical Measurement Techniques IPM and the Max Planck Institute for Dynamics and Self-Organization (MPI-DS). Because high-resolution measurements are best achieved at short measurement ranges, the CTL system is designed to be mounted on an airborne platform such as the Max Planck CloudKite (MPCK) (Bagheri et al., 2018; Schröder et al., 2021; Stevens et al., 2021), an instrumented balloon–kite hybrid capable of flying up to 2 km above the ground. The CTL is based on a multi-beam arrangement and uses an FMCW (frequency-modulated continuous-wave) laser to measure wind speeds in the vicinity of the carrier platform, e.g., at a distance of 10–15 m. With this approach, non-intrusive high-resolution measurements can be achieved. The 3D wind vector is resolved by focusing three independent, spatially separated line-of-sight lidar measurement channels on one single measuring point.

The CTL enables single-point 3D wind measurements at the meter scale with a temporal resolution of typically 10 Hz and thus opens up the possibility of investigating turbulence on a much smaller scale than with classic scanning lidars (Pauscher et al., 2016) and at altitudes and in situations which were previously inaccessible. Further use cases with limited space like wind turbines or meteorological masts are conceivable. However, there are systematic constraints when using a lidar system on an airborne platform. The main measurement uncertainty results from the individual measurement errors and the limited space for mounting (see Appendix A for the consideration of other error sources, such as effects of temperature and platform motion). The small distance between the telescopes from which the laser beams originate means that the three lidar beams have a large angle to the transverse components. The measurement uncertainty of the resulting reconstructed 3D wind vector can be calculated by error propagation theory from the intrinsic measurement uncertainty and the system geometry. Due to the small angles, the spatial dimensions which are transverse to the main direction of the system suffer from high uncertainties. This might constrain the use cases of the CTL for its application. However, as shown in this study, considering this effect in post-processing can enhance the data quality.

In this study, the measurement uncertainties are analyzed and an uncertainty propagation model is introduced to identify the dependencies of uncertainty, geometry, measurement noise model, and turbulence characteristics of the measured wind data. This is done using synthetic wind data generated by an Ornstein–Uhlenbeck process, a well-known model for simulating turbulent wind data (Uhlenbeck and Ornstein, 1930; Pope, 2011; Zárate-Miñano et al., 2013). Our analysis shows that it is possible to reduce the uncertainties of the transverse components of a measured 3D wind vector by applying appropriate low-pass filtering to the data, i.e., averaging of the data points.

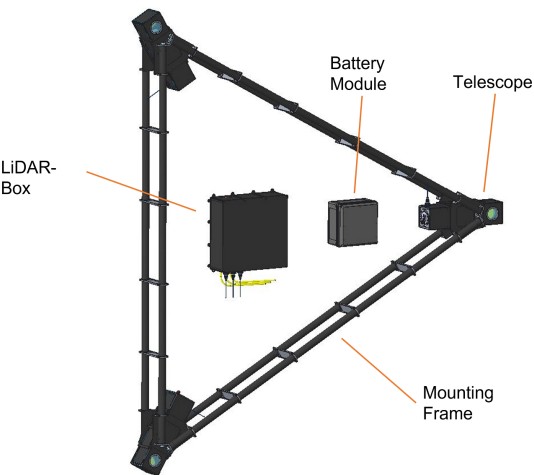

**Figure 1.** Schematic of the CloudKite Turbulence LiDAR (CTL). It consists of three optical heads (telescopes): the lidar box including data processing and control module, the battery module, and the carbon mounting frame. The side length of the triangle is $d_t = 3$ m.

**Table 1.** Specifications of the CloudKite Turbulence LiDAR (CTL).

| | |
|---|---|
| Measurement rate | 10 Hz |
| Spatial resolution | $\leq 1\,\mathrm{m}^3$ |
| Spatial dimensions | 3 |
| Wind velocity accuracy (LoS) | $< 0.1\,\mathrm{m\,s}^{-1}$ |
| Measuring distance | 7–50 m |
| Laser wavelength | 1545 nm |

Furthermore, a series of analytical expressions are developed to determine the best post-processing parameters that minimize the measurement uncertainty of a given wind data set and also to illustrate how they can be applied to real infield data. The latter is done by using experimental wind data taken with the MPCK in the framework of the EUREC[4]A campaign (Stevens et al., 2021). Our results highlight the reliability and potential of CTL for use in future field campaigns to characterize ABLs at high resolution while providing the necessary post-processing tools for analysis of the collected data by CTL and systems of similar design concepts.

## 2 Setup description

### 2.1 The CloudKite Turbulence LiDAR

The setup of the novel, currently developed 3D wind lidar is shown in Fig. 1. The main specifications are summarized in Table 1. The core optical lidar module with three optical channels was custom-built by ABACUS Laser GmbH (Göttingen, Germany) based on a joint concept development.

The core lidar was integrated into a system design that spatially separates the three optical heads, which are equipped

with focusing 3 in. (7.62 cm) CE1 telescopes. The laser beams are focused on one measurement volume, which ensures a spatial resolution of $\leq 1\,\mathrm{m}^3$. This configuration allows reconstructing a 3D wind vector from the three line-of-sight (LoS) measurements. The separation of the laser telescopes and the focus distance is chosen to maximize the accuracy of the wind vector measurement within the limitations of the mounting platform.

For the measurement of the three LoS wind velocities an FMCW scheme is used. A single continuous-wave laser source is split into three measurement channels and a reference channel. Each measurement channel is connected to one telescope. An additional reference channel consists of an internal glass fiber of fixed length. The laser source is frequency-modulated using a triangular function with, typically, a 10 kHz base frequency. Detection of the signal is done using a balanced photodetector for each channel, where the backscattered signal interferes with a part of the laser source (local oscillator, LO). The power of the LO is adjusted so that the detection operates in the shot-noise-limited regime. That means that shot noise is the dominant noise source in the detection path.

For extraction of a wind velocity, data analysis is typically done as follows (same for each channel): the raw data from one modulation period, corresponding to one rising and one falling slope of the laser frequency, are divided into 12 equally long sections (each 8.3 µs). Each section is multiplied with a flat-top window and then fast-Fourier-transformed to yield a frequency spectrum. A total of 1000 subsequent spectra of identical sections are then averaged. The precise peak position in these spectra is estimated by applying a Gaussian fit. The peak frequencies of the four central sections of the rising edge are averaged to yield $F^+$. The same is done for the falling edge to yield $F^-$. The wind velocity is then given by $v_{\mathrm{wind}} = (F^- - F^+)/2$. Under typical atmospheric conditions, a velocity resolution can be achieved of at least $0.1\,\mathrm{m\,s^{-1}}$, with a temporal resolution of 10 Hz, according to the specifications provided by the manufacturer.

The usage of the CTL on the MPCK enables short-range remote wind measurements with high lateral spatial resolution and high velocity resolution of the transverse spatial components at altitudes of interest within the atmospheric boundary layer.

## 2.2 Measurement uncertainty

We TS1 assume that every measurement in each of the three channels experiences a statistically independent, normally distributed error. The standard deviation of this error distribution shall be called the measurement uncertainty $\sigma^{\mathrm{det}}$. The manufacturer of the wind lidar module provides an estimate of the measurement accuracy in terms of a full-width half-maximum (FWHM) of $0.1\,\mathrm{m\,s^{-1}}$, derived from the fluctuations of the velocity value when measuring a constant wind value. This is comparable to other values provided in the literature (Knoop et al., 2021). As $\sigma^{\mathrm{det}} = \mathrm{FWHM}/(2\sqrt{2\ln 2})$, a measurement uncertainty in terms of standard deviation can conservatively be estimated to be $\sigma^{\mathrm{det}} = 0.04\,\mathrm{m\,s^{-1}}$.

To further justify these assumptions, various experiments and considerations were done. First, data from the reference channel and from measurements on a hard target (lab wall) were analyzed. These are scenarios in which the signal-to-noise ratio (SNR) is much higher than in any wind measurement. Consequently, they show the limitations of the measurement system in the case of high SNR. In both cases, the distribution of the velocity measurements approximates a normal distribution (see Appendix B). The derived measurement uncertainty is $\sigma^{\mathrm{det}} = 9 \times 10^{-5}\,\mathrm{m\,s^{-1}}$ (reference channel) and $\sigma^{\mathrm{det}} = 1.2 \times 10^{-4}\,\mathrm{m\,s^{-1}}$ (hard target).

As the detection is shot-noise-limited, the effects of low SNR can be simulated. As shown in Appendix C, the measurement uncertainty increases with decreasing SNR and remains normally distributed, even for very low SNRs, which are comparable to low aerosol densities. In this low-SNR regime, the simulated measurement uncertainty is $\sigma^{\mathrm{det}} \approx 0.01\,\mathrm{m\,s^{-1}}$. This indicates that the magnitude of $\sigma^{\mathrm{det}} = 0.04\,\mathrm{m\,s^{-1}}$ is a valid, conservative assumption.

For real wind measurements the fluctuations of the velocity will likely be greater. Even within 100 ms the wind speed is typically neither constant nor homogeneous within the whole focal volume. However, this is not an uncertainty due to the measurement but rather an intrinsic property of the quantity under observation.

## 2.3 Measurement geometry

Figure 2 shows the geometry of the MPCK and the CTL with its three telescopes mounted on the keel of the MPCK kite. The global coordinate system $x$–$y$–$z$ is defined as shown in Fig. 2a, where the MPCK's keel tail end is pointing in the $x$ direction and usually aligns with the direction of the mean wind. The lidar measurement geometry constitutes a pyramid with an equilateral triangle as the base, a telescope at each corner, and the focus point at the top edge (see Fig. 2b). The distance $d_{\mathrm{t}}$ between two telescopes defines the side length of the base, and the length of one long edge is defined by the focus distance $d_{\mathrm{f}}$. The height of the pyramid is denoted by $h$, which corresponds to the distance of the mounting platform to the focus point. The unit vectors in the line-of-sight direction of the three measurement channels are defined in the measurement coordinate system $u$–$v$–$w$ as

$$\hat{\boldsymbol{u}}_1 = \begin{bmatrix} \cos\theta \\ 0 \\ \sin\theta \end{bmatrix}, \quad \hat{\boldsymbol{u}}_2 = \begin{bmatrix} -\frac{1}{2}\cos\theta \\ \frac{\sqrt{3}}{2}\cos\theta \\ \sin\theta \end{bmatrix},$$

$$\hat{\boldsymbol{u}}_3 = \begin{bmatrix} -\frac{1}{2}\cos\theta \\ \frac{-\sqrt{3}}{2}\cos\theta \\ \sin\theta \end{bmatrix}, \tag{1}$$

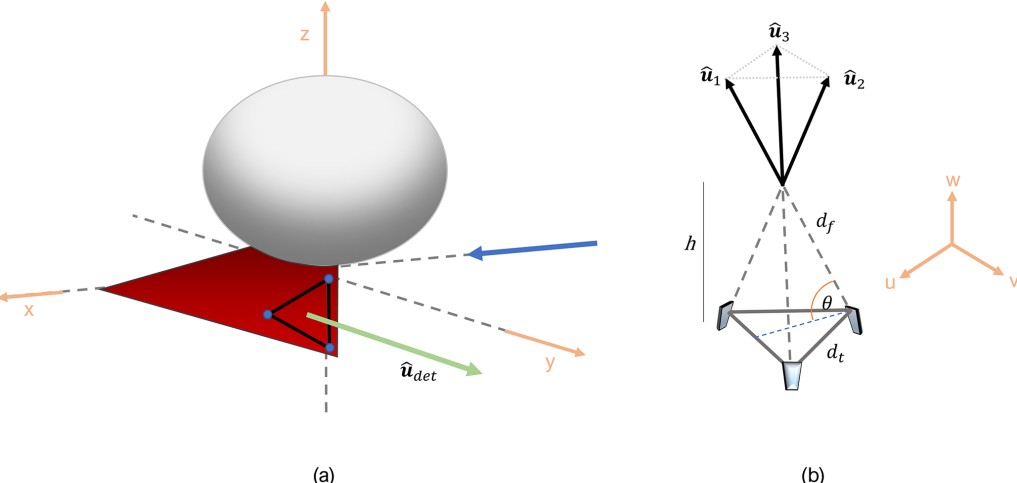

**Figure 2. (a)** Schematic of the MPCK in the global coordinate system $x-y-z$. The MPCK aligns with the mean wind direction (blue arrow, $x$ axis). $\hat{u}_{\text{det}}$ points in the main direction of the measurement system. **(b)** Geometry of the detection system in the measurement coordinate system $u-v-w$. The direction of the detection system $\hat{u}_{\text{det}}$ **(a)** is perpendicular to the base of the detection system and points in the direction of the spatial component $w$. This component is denoted as the longitudinal component of the measured wind data. The lateral and vertical components $u$ and $v$ are denoted as transverse components. $d_{\text{t}}$ and $d_{\text{f}}$ are the spatial distance of the optical telescopes and their focus distance, respectively. The distance of the lidar ground plane to the measuring volume is denoted as $h$. $\hat{u}_1$, $\hat{u}_2$, and $\hat{u}_3$ are the unit vectors in the line-of-sight direction of each lidar measurement channel (optical head).

with angle $\theta = \arccos \frac{d_{\text{t}}}{\sqrt{3}d_{\text{f}}}$. The direction of the entire measurement system $\hat{u}_{\text{det}}$ is defined in the measurement coordinate system $u-v-w$ as the direction of the $w$ axis:

$$\hat{u}_{\text{det}} = \begin{bmatrix} 0 \\ 0 \\ 1 \end{bmatrix}. \tag{2}$$

The measurement coordinate system $(u-v-w)$ is defined by a rotation relative to the global coordinate system $(x-y-z)$. A vector $a'$ in the measurement coordinate system is described by a vector $a$ in the global coordinate system as

$$a' = \mathbf{R}_x(\alpha)\,\mathbf{R}_y(\beta)\,\mathbf{R}_z(\gamma)\,a. \tag{3}$$

The rotation matrices $\mathbf{R}_x(\alpha)$, $\mathbf{R}_y(\beta)$, and $\mathbf{R}_z(\gamma)$ describe the counterclockwise rotation of a vector by a certain angle about the given axis. For the analysis and the results presented in this paper, we first rotated the system around the $z$ axis by the angle $\gamma$, which corresponds to an intrinsic rotation around the $w$ axis. This is followed by a rotation by the angle $\beta$ around the $y$ axis and then by the angle $\alpha$ around the $x$ axis. With $\alpha = 90°$, $\beta = 0°$, and $\gamma = 0°$ the measurement system points in the $y$ direction, which is transverse to the mean wind direction.

The unit vectors of the measurement system $\hat{u}_{\text{det}}$, $\hat{u}_1$, $\hat{u}_2$, and $\hat{u}_3$ are defined in the measurement coordinate system (Eqs. 1 and 2) and can be transferred to the global coordinate system by Eq. (3). In the following all vectors and matrices are defined in the global coordinate system. With the rotations defined above ($\alpha = 90°$, $\beta = 0°$, and $\gamma = 0°$) the longitudinal component of the measurement system ($w$) aligns

with the $y$ component and the transverse components $(u, v)$ with the $x$ and $z$ components of the global measurement system.

For the results of this paper, the distance between two telescopes is assumed to be $d_{\text{t}} = 3$ m and the focus distance is set to $d_{\text{f}} = 15$ m. In this focus distance, it is assumed to measure wind which is not affected by wind turbulence introduced by the lidar mounting platform, i.e., the MPCK.

## 3 Methods

### 3.1 Synthetic wind data

We want to model wind data without wind gusts or large changes in atmospheric conditions and relatively weak turbulence intensity. For this kind of wind, the single-point velocity probability density function (PDF) can be assumed to be Gaussian (Calif, 2012). For generating such synthetic wind fluctuation data, an Ornstein–Uhlenbeck (OU) process can be used as a simple stochastic differential equation (SDE) model (Zárate-Miñano et al., 2013; Pope, 2011; Risken, 1989), which allows controlling fluctuations, time correlations, and turbulence intensity.

According to an OU process, each component of the synthetic wind vector $v_i^{\text{sim}}(t)$ evolves over a time step $\mathrm{d}t$ as follows:

$$\mathrm{d}v_i = -\frac{1}{\tau}[v_i - \mu_i]\mathrm{d}t + \sqrt{\frac{2\mathrm{Var}}{\tau}}\,\mathrm{d}W_i. \tag{4}$$

Here, the first term on the right-hand side corresponds to a drift term toward the mean wind velocity $\mu_i$, with the index $i$ referring to the spatial coordinates $x$, $y$, and $z$. The second term is a stochastic term featuring the increment $\mathrm{d}W_i$ of a Wiener process. The parameters $\tau > 0$ and $\text{Var} > 0$ are the correlation time and the variance of the generated wind data, respectively. To generate a synthetic wind data set, the OU process is discretized using the Euler–Maruyama method (Kloeden and Platen, 1992) and implemented in Python code.

For the uncertainty propagation model, a synthetic wind data set is needed with realistic and typical turbulence characteristics. As typical values we consider the variance (Var) to be of the order of $1$–$4\,\mathrm{m^2\,s^{-2}}$ and the correlation time $\tau$ to be $5$–$10\,\mathrm{s}$. The characterization of the experimental data (Sect. 3.2), which will be used later in this work, exhibits values for the variance ranging from $0.02$ up to $5.2\,\mathrm{m^2\,s^{-2}}$. This broad distribution makes it difficult to choose one "typical" value of variance for the synthetic wind data set. For the correlation time it is challenging to derive accurate values from the experimental wind data available, as discussed later (see Sect. 6).

Based on these considerations a synthetic data set $\boldsymbol{v}^{\text{sim}}$ was used, with variance of the data set of $\text{Var} = 1\,\mathrm{m^2\,s^{-2}}$ and a correlation time of $\tau = 7.5\,\mathrm{s}$. For simplicity, the same values are chosen for each spatial component $v_i^{\text{sim}}$. The mean velocity components are chosen as $\mu_x = 8\,\mathrm{m\,s^{-1}}$, $\mu_y = -4\,\mathrm{m\,s^{-1}}$, and $\mu_z = 0\,\mathrm{m\,s^{-1}}$. **TS2**

## 3.2 Experimental wind data

The experimental wind velocity data $\boldsymbol{v}^{\text{exp}}$ used for the analysis were acquired by a 3D ultrasonic anemometer mounted on the MPCK, which measured all three spatial components with a 30 Hz measurement frequency. The data are samples of 1–2 h duration taken during flights at different altitudes as part of the EUREC[4]A campaign (https://eurec4a.eu/, last access: 20 November 2024) on RV *Meteor* (Stevens et al., 2021).

In addition to the MPCK data, ground-based measured wind data have been used for the present investigation. The data were taken by Augustinus Cornelis Maria Bertens from MPI-DS with a 3D ultrasonic anemometer (Ultrasonic Anemometer3D; part no. 4.3830.20.340; Thies Clima, Göttingen) at the research station Schneefernerhaus close to Zugspitze (Bertens, 2021).

To get measurement data with characteristics as similar as possible to a data set of the CTL, the time resolution of the experimental data set has to be reduced to $\Delta t = 0.1\,\mathrm{s}$. This is achieved by merging consecutive data points by an arithmetic average.

The experimental wind data are characterized using mean velocity, variance, and correlation time. The mean velocity is defined as $\overline{v} = \frac{1}{N}\sum_{t=0}^{N} v_t$ and the variance as $\sigma_v^2 = \frac{1}{N-1}\sum_{t=0}^{N}(v_t - \overline{v})$, with $N$ as the number of data points.

The correlation time $T$ of the time series can be calculated by computing the integral of the normalized autocorrelation function or from a fit of the function $C(t) = \exp(-t/\tau)$ to the autocorrelation data. Based on the same exponential assumption, another approach to calculating the correlation time is $\tau = -1/\ln(C(1))$. The methods could be validated by applying them to synthetic data and comparing the correlation time calculated by the Ornstein–Uhlenbeck parameter with the result from the autocorrelation function. For the results presented in the following sections, the latter approach was chosen.

## 3.3 Uncertainty propagation model

This section presents the uncertainty propagation model. It takes, as an input a wind data set, either synthetic or experimental, and calculates the expected measurement data by projecting the wind data on the directions of the measurement channels. Then, statistically independent Gaussian-distributed deviations are added to each measurement channel data point, which is meant to simulate the intrinsic measurement uncertainty. The "erroneous" measurement data are then reconstructed and compared to the input data set. This uncertainty analysis reveals the dependencies of the measurement uncertainty for a multi-beam wind lidar.

An initial wind speed vector is denoted as $\boldsymbol{v}^{\text{init}}(t)$ and is provided either from a theoretical turbulence model, i.e., a synthetic data set $\boldsymbol{v}^{\text{sim}}(t)$ (see Sect. 3.1), or from field measurements $\boldsymbol{v}^{\text{exp}}(t)$ (see Sect. 3.2). The expected measurement data $v_d^{\text{det}}(t)$ for each lidar channel $d$ are the line-of-sight components in the direction of the measurement unit vectors $\hat{\boldsymbol{u}}_d(t)$ (see Eq. 1) with $d = 1, 2$, and 3 and can be calculated by projecting the initial wind vector $\boldsymbol{v}^{\text{init}}(t)$ on the measurement unit vectors:

$$v_d^{\text{det}}(t) = \hat{\boldsymbol{u}}_d \cdot \boldsymbol{v}^{\text{init}}(t). \tag{5}$$

The values of all three measurement channels form a vector $\boldsymbol{v}^{\text{det}}(t)$. In this notation Eq. (5) becomes

$$\boldsymbol{v}^{\text{det}}(t) = \mathbf{M}^T \boldsymbol{v}^{\text{init}}(t), \tag{6}$$

with

$$\boldsymbol{v}^{\text{det}} = \left(v_1^{\text{det}}, v_2^{\text{det}}, v_3^{\text{det}}\right)$$

and

$$\mathbf{M} = \left(\hat{\boldsymbol{u}}_1, \hat{\boldsymbol{u}}_2, \hat{\boldsymbol{u}}_3\right).$$

Each measurement channel has a certain intrinsic measurement uncertainty. As there is no precise knowledge about the origin of the measurement uncertainty, we model realistic measurement data by adding a random deviation to each measurement channel for each time step, which can be regarded as simulated errors. This has been done similarly by Schlipf et al. (2020). The deviations or errors $\delta_d^{\text{det}}$ are

**Table 2.** Parameter choices for all figures and results unless otherwise stated (default configuration).

| | | |
|---|---|---|
| Telescope distance | $d_t$ | 3 m |
| Focus distance | $d_f$ | 15 m |
| Direction of measurement system | $\hat{\boldsymbol{u}}_{det}$ | $y$ |
| Measurement uncertainty | $\sigma^{det}$ | $0.04 \text{ m s}^{-1}$ |
| Measurement rate | $f_s$ | 10 Hz |
| Variance of synthetic data | Var | $1 \text{ m}^2 \text{ s}^{-2}$ |
| Correlation time of synthetic data | $\tau$ | 7.5 s |
| Mean reversion levels | $\mu_x$ | $8 \text{ m s}^{-1}$ |
| | $\mu_y$ | $-4 \text{ m s}^{-1}$ TS3 |
| | $\mu_z$ | $0 \text{ m s}^{-1}$ |

Gaussian-distributed with zero mean and a standard deviation $\sigma_d^{det}$ for each measurement channel $d$, estimated by assumptions on the measurement principle and the initial configuration of the system (see Table 2). $\sigma^{det}$ is denoted as the "intrinsic measurement uncertainty" of the measurement system and is assumed to be the same for all measurement channels. The wind data for the measurement channel $d$ with added error are denoted as $v_d^{err}$ and are defined by

$$v_d^{err}(t) = v_d^{det}(t) + \delta_d^{det}(t). \tag{7}$$

In an actual experiment, there is no a priori knowledge of $\boldsymbol{v}^{init}(t)$ or $\boldsymbol{v}^{det}(t)$. Only the "erroneous" measurement data $\boldsymbol{v}^{err}(t)$ are available. Using widely used reconstruction formulas (e.g., Holtom and Brooms, 2020; Schlipf et al., 2012, 2020) and applying the geometry of the measurement system, it is possible to reconstruct a 3D wind vector $\boldsymbol{v}^{recon}(t)$ from the erroneous measurement data as

$$\boldsymbol{v}^{recon}(t) = \left(\mathbf{M}^T\right)^{-1} \boldsymbol{v}^{err}(t) = \mathbf{T}\boldsymbol{v}^{err}(t), \tag{8}$$

where $\mathbf{T} = (\mathbf{M}^T)^{-1}$ denotes the reconstruction matrix. The result of this reconstruction algorithm is a 3D wind vector with an intrinsic measurement uncertainty.

## 3.4 Post-processing of reconstructed wind data

The lidar measurement channels introduce errors that can be regarded as statistically independent. The fluctuation of the data due to wind turbulence is, however, correlated between all three channels. Because of this, applying a post-processing averaging to the resulting reconstructed data might be advantageous to reduce the resulting measurement uncertainty of the reconstructed 3D wind vectors. This will not reduce the number of time steps but will smooth out fluctuations on the scale of the averaging time, which can be interpreted as reducing the "physical" time resolution. The aim of this post-processing is to reduce the measurement un-

certainty but not lose information about relevant turbulence characteristics in the data.

The post-processing averaging can be implemented as a low-pass filtering of the respective component of the reconstructed wind velocity vector. Different approaches are discussed and compared in Appendix D. For the present investigations, a Gaussian filter was chosen as an implementation of low-pass filtering. It can be interpreted as a moving average with Gaussian weights. The filter function is defined with the standard deviation $\sigma^{filt}$ as

$$g(t) = \frac{1}{\sqrt{2\pi}\sigma^{filt}} \exp\left(\frac{-t^2}{2(\sigma^{filt})^2}\right). \tag{9}$$

The filtering is done by convolving the data set with the filter function. Here, the given Gaussian filter function is truncated to a window function with the length of $4\sigma^{filt}$. To simplify the interpretation of the results of the analysis with Gaussian filtering, the standard deviation of the Gaussian filter is set to $\sigma^{filt} = n/4$, where $n$ is the length of the window. Using a simple moving average with $n$ as the number of averaged data points for the analysis yields similar results, i.e., the same minima of uncertainty as $n$ changes (see Appendix D).

## 3.5 Evaluation of processed data

The measurement uncertainty of the reconstructed wind data is determined by comparing each time step of the initial wind velocity $\boldsymbol{v}^{init}(t)$ with the reconstructed and post-processed wind data $\boldsymbol{v}^{recon}(t)$. The deviations of both data sets at a time $t$ are defined for each spatial component $i$ as

$$a_{i,t} = v_{i,t}^{init} - v_{i,t}^{recon}. \tag{10}$$

The measurement uncertainty $\sigma_i$ TS4 is calculated for each spatial component $i$ in terms of the standard deviations of the distribution of the pointwise deviations $a_{i,t}$ of both data sets:

$$\sigma_i = \sqrt{\frac{1}{N}\sum_t^N (a_{i,t} - \mu_i)^2}, \tag{11}$$

with

$$\mu_i = \frac{1}{N}\sum_t^N a_{i,t}.$$

## 3.6 Error propagation theory

The error propagation theory describes how uncertainties or random errors of a function depend on the uncertainties of variables in the function definition. The theory describes the variables of the functions as experimental quantities that have a certain uncertainty due to measurement limitations. For the following analysis this theory is used to compare the results of the uncertainty propagation model with theoretical values and justify our approach.

If the function is a linear combination $f = \sum_j^n a_j x_j$ and in the case of uncorrelated variables, the uncertainty of the function $\sigma^f$ with variables $x_j$, coefficients $a_j$, and uncertainty of the variables $\sigma_j$ is defined as (Joint Committee for Guides in Metrology, 2008)

$$(\sigma^f)^2 = \sum_j^n \sigma_j^2 a_j^2. \tag{12}$$

Each spatial component of the reconstructed 3D wind vector $\boldsymbol{v}_i^{\text{recon}}$ of the 3D FMCW wind lidar in the Cartesian coordinate system is a linear combination of the measurement data $\boldsymbol{v}_1^{\text{err}}$, $\boldsymbol{v}_2^{\text{err}}$, and $\boldsymbol{v}_3^{\text{err}}$ (see Sect. 3.3):

$$v_i^{\text{recon}} = \mathbf{T}^{i1} \boldsymbol{v}_1^{\text{err}} + \mathbf{T}^{i2} \boldsymbol{v}_2^{\text{err}} + \mathbf{T}^{i3} \boldsymbol{v}_3^{\text{err}}. \tag{13}$$

$\mathbf{T}$ denotes the reconstruction matrix, defined in Sect. 3.3, and only depends on the geometrical constraints of the measurement system. From this and Eq. (12), it follows for the theoretical uncertainty of the spatial components of the reconstructed wind vector $\sigma_i^{\text{theory}}$, with $\sigma^{\text{det}}$ as the intrinsic measurement uncertainty (which is assumed to be the same for all measurement channels), that TS5

$$\sigma_i^{\text{theory}} = \sqrt{\left(\mathbf{T}^{i1}\right)^2 + \left(\mathbf{T}^{i2}\right)^2 + \left(\mathbf{T}^{i3}\right)^2} \, \sigma^{\text{det}}. \tag{14}$$

## 4 Theoretical analysis

### 4.1 Assumptions

The results of the following analysis are based on assumptions on the measurement geometry, turbulence characteristics of the synthetic wind data set, and the measurement uncertainty. All parameter choices are summarized in Table 2 and are used for all figures and results (denoted as the default configuration of the system) unless otherwise stated. Due to reasons of radial symmetry, the uncertainty propagation model gives the same results for the transverse components $x$ and $z$ when using synthetic wind data. Therefore, only the results of the transverse component $x$ are presented for the part with synthetic data.

### 4.2 Theoretical uncertainty propagation

A theoretical approach to calculate the uncertainty of the reconstructed wind vector is the error propagation theory, which is introduced in Sect. 3.6. Equation (14) defines the theoretical uncertainty of the spatial components of the reconstructed wind vector $v_i^{\text{recon}}$ in relation to the reconstruction matrix $\mathbf{T}$ and the intrinsic measurement uncertainty $\sigma^{\text{det}}$. With the input parameters as given in Table 2, the theoretical uncertainty of the spatial components of the reconstructed wind vector is $\sigma_x^{\text{theory}}, \sigma_z^{\text{theory}} = 0.28 \, \text{m s}^{-1}$ for the transverse components, and $\sigma_y^{\text{theory}} = 0.02 \, \text{m s}^{-1}$ for the longitudinal component in direction of the measurement system.

### 4.3 Uncertainty analysis without post-processing

This section and the next sections present the results of the uncertainty propagation analysis. With the uncertainty propagation model described in Sect. 3.3 the measurement uncertainty of a three-beam wind lidar like the CTL can be estimated based on geometric constraints, turbulence characteristics, and post-processing averaging. A synthetic wind data set with defined turbulence characteristics is used as an input data set (see Sect. 3.1) and the expected measurement data of each respective measurement channel are calculated (Eq. 5). After adding random Gaussian errors (Eq. 7), which simulate the intrinsic measurement uncertainty, a reconstruction algorithm is applied (Eq. 8), and the resulting data set is compared with the input data (Eq. 10). The measurement uncertainty of each component is calculated as the standard deviation of the distribution of the pointwise deviations of the two data sets (Eq. 11). The following results are based on the default configuration of the system as defined in Table 2.

Figure 3 shows the resulting measurement uncertainty of the CTL if no post-processing is applied to the reconstructed measurement data. The plots show the comparison of the input data set (synthetic wind data) and the reconstructed data set. As expected, the values of the measurement uncertainty are the same as calculated by error propagation theory (see Sect. 4.2). This shows that in the simple case of reconstructing and analyzing the data for each time step individually, the overall measurement uncertainty depends only on the input measurement uncertainty and geometrical parameters. The characteristics of the input data, i.e., the fluctuation of the data and the mean values, do not influence the result in the case of not applying post-process averaging.

### 4.4 Uncertainty analysis with post-processing

Figure 4 shows the results of the uncertainty propagation analysis when applying a post-processing to the reconstructed measurement data. The same input parameters are used as for the results in Fig. 3. The measurement uncertainty of the $x$ component (transverse component) is $\sigma_{x,n=1} = 0.28 \, \text{m s}^{-1}$ without post-processing and $\sigma_{x,n=6} = 0.15 \, \text{m s}^{-1}$ when applying a Gaussian filter with a window length of six data points (as explained in Sect. 3.4). The uncertainties for the $y$ component (longitudinal component) are $\sigma_{y,n=1} = 0.023 \, \text{m s}^{-1}$ without averaging and $\sigma_{y,n=6} = 0.09 \, \text{m s}^{-1}$ in the post-processed case.

Figure 5 shows the behavior of the measurement uncertainty depending on the filter length, i.e., averaging time for the relevant spatial components. The figure shows that in the case of the transverse component $(x)$ the uncertainty is reduced and reaches a minimum at around seven data points. This reduction of the measurement uncertainty comes at the cost of increasing the measurement uncertainty of the longitudinal component. Nevertheless, the results show that there is a clear benefit of applying a post-processing averaging

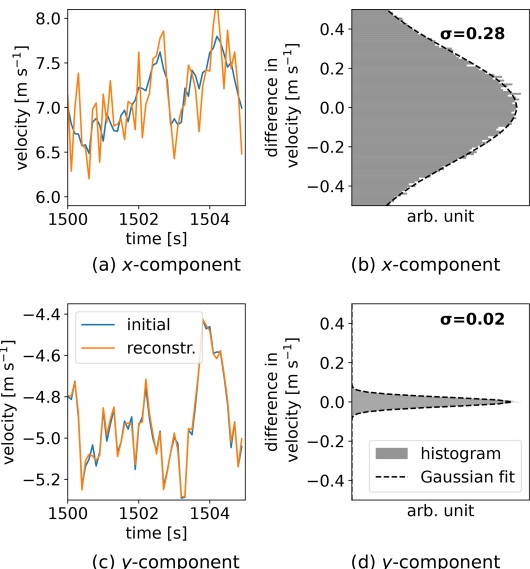

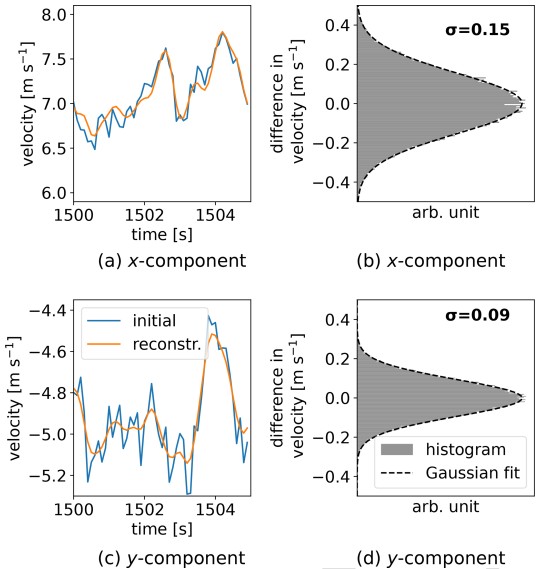

**Figure 3.** Results of the measurement uncertainty propagation model without applying a post-processing averaging. This figure compares the initial data set with the "erroneous", reconstructed data set. The uncertainty of the transverse component ($x$ component) **(a, b)** is increased compared to the measurement uncertainty of $\sigma^{\mathrm{det}} = 0.04\,\mathrm{m\,s^{-1}}$. The longitudinal component ($y$ component) **(c, d)** shows a reduced uncertainty. **(a, c)** Segment of the input data (blue) and the corresponding reconstructed data (orange). **(b, d)** Normalized histogram of the deviation between input and reconstructed data. The dashed line shows a Gaussian fit to the distribution. The measurement uncertainty $\sigma$ resulting from this fit is given in the inset. TS6

to the reconstructed wind data with lengths of up to seven data points. In this range, the measurement uncertainty of the transverse component is significantly reduced, while the longitudinal uncertainty still remains below the uncertainty of the transverse component. Another possibility is to only apply the post-processing to the transverse component. This approach does not increase the uncertainty of the longitudinal component. However, it depends on the application of the data whether a differentiated processing of the individual wind data components is permissible or not.

## 4.5  Dependence on the measurement rate

Up to now, we have considered the uncertainty propagation model with synthetic input data with a fixed measurement rate. The CTL is developed to measure with a rate of 10 Hz. However, depending on the aerosol density in the air, higher (or lower) measurement rates are possible, while preserving the same SNR in the frequency spectra from which the wind velocities are extracted. Thus, it is worth investigating how the post-processing parameters for decreasing the measurement uncertainty depend on the measurement rate of the system. Figure 6 shows the results of the uncer-

**Figure 4.** Uncertainty propagation analysis with post-processing. A Gaussian filter is applied with a window length of $n = 6$ data points. **(a, c)** Segment of the input wind velocity data (blue) and the corresponding reconstructed data (orange). **(b, d)** Normalized histogram of the deviation between the input data and the reconstructed and post-processed data. The dashed line shows a Gaussian fit to the distribution. The measurement uncertainty $\sigma$ is given in the text insert. **(a, b)** Transverse component ($x$ component). **(c, d)** Longitudinal component ($y$ component).

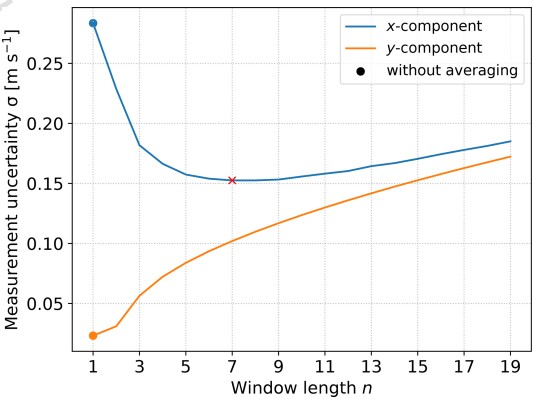

**Figure 5.** The measurement uncertainty depends on the length of the post-processing averaging window $n$. The plot shows the results of the uncertainty propagation analysis based on a synthetic wind data set with input parameters as defined in Table 2. The $x$ component corresponds to a transverse component of the wind vector in the measurement coordinate system and the $y$ component denotes the longitudinal component. The curve of the measurement uncertainty of the transverse component has a minimum for a certain averaging length, in this specific case at about seven data points (red cross).

tainty propagation model in the default configuration (see Table 2) with synthetic input data with various measurement

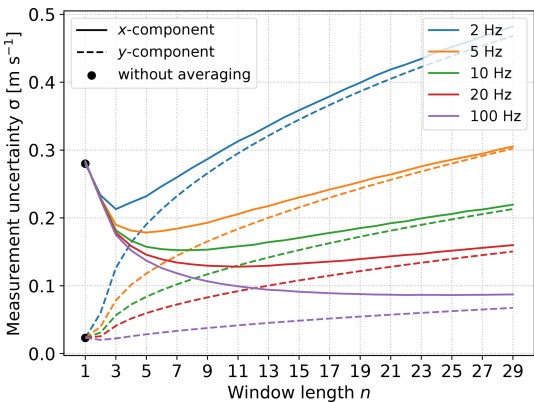

**Figure 6.** Measurement uncertainty of reconstructed and post-processed wind components for various measurement rates. Results are based on the uncertainty propagation model with a synthetic wind data set in default configuration (see Table 2).

rates. To generate such data, the time step of the Ornstein–Uhlenbeck process is changed while keeping the other input parameters (Var, $\tau$, and $\mu_i$) constant. The results show that the uncertainty of the transverse component is smaller for higher measurement rates independent of the window length (Fig. 6). Furthermore, the minimum of the transverse uncertainty shifts to larger window lengths $n$ for higher measurement rates. At lower measurement rates, the ability to reduce the transverse uncertainty becomes smaller. It thus follows that, compared to the results of the previous sections, the transverse uncertainty can be even further reduced when the aerosol density in the air allows for higher measurement rates.

From a physical point of view, it would also make sense to investigate various measurement rates depending on an averaging time instead of the window length $n$. However, for the experimental setup and its application in the field, the window length is the relevant quantity and was thus chosen as the variable parameter.

### 4.6 Dependence of the uncertainty on turbulence characteristics

As mentioned above, the ability to reduce the measurement uncertainty by averaging over multiple data points depends on geometric parameters and the measurement uncertainty on the one hand and turbulence characteristics on the other hand. Increasing the averaging length will first decrease the uncertainty of the transverse components until a minimum is reached (Fig. 5). This minimum depends on the turbulence characteristics, i.e., the size (variance) and integral time length (correlation time) of turbulent fluctuations of the data set. Figure 7a shows the average lengths that yield the minimum measurement uncertainty of the transverse components for a wide range of typical turbulence characteristics. Figure 7b gives the value of the respective uncertainty minima.

The results plotted in the figures are calculated as follows: for a given variance and correlation time, a synthetic wind data set is generated. The uncertainty propagation model provides the dependency of the measurement uncertainty on the averaging length, i.e., the length of the filter window. The window length for which the uncertainty gets minimized is determined and plotted for various values of variance and correlation time. Input parameters for Fig. 7 are the same as above and noted in Table 2.

The results show that the measurement uncertainty of the transverse component can be reduced compared to the case without averaging ($\sigma_{x,n=1} = 0.28\,\mathrm{m\,s^{-1}}$) for all turbulence values used for the calculations. In the case of small variance and long correlation time, it gets reduced the most. In this case, the weak fluctuation of the data allows long averaging without losing measurement accuracy.

## 5 Experimental application

The results of the last section (Sect. 4) are based on synthetic data generated by an Ornstein–Uhlenbeck process. The dependency of the measurement uncertainty of the reconstructed wind components on the length of the averaging window, turbulence characteristics of the wind data set, and other parameters was analyzed. In this section, experimental input data sets are used as input data for the uncertainty propagation model. It will be investigated whether the findings from the first part can be transferred to an actual experiment.

### 5.1 Uncertainty analysis with experimental wind data

For various experimental data sets the measurement uncertainty of the CTL in default configuration (see Table 2) is calculated depending on the length of the averaging window as explained in Sect. 3.3. For this, the experimental wind data are taken as the initial wind data, assuming that the data set represents the actual wind for the measurement rate used. Then the measurement data are calculated, which means a projection of the initial data on the measurement unit vectors. After adding Gaussian-distributed errors at each measurement channel, the 3D data set is reconstructed and compared to the initial data set. The results of the uncertainty analysis with experimental data (see Sect. 3.2) from the MPCK and from ground-based sonic anemometer measurements are plotted in Fig. 8 (only the transverse component $x$ is shown). The uncertainty of the transverse component ($x$) is reduced for all data sets and for averaging lengths up to nine data points. The $z$ component shows similar behavior. The longitudinal component ($y$) increases for all averaging lengths, which is not shown in the figures. It is possible to approximately halve the measurement uncertainty of the transverse components. In this case, the uncertainty of the longitudinal component increases but stays below the uncertainty of the transverse component. In conclusion, by using experi-

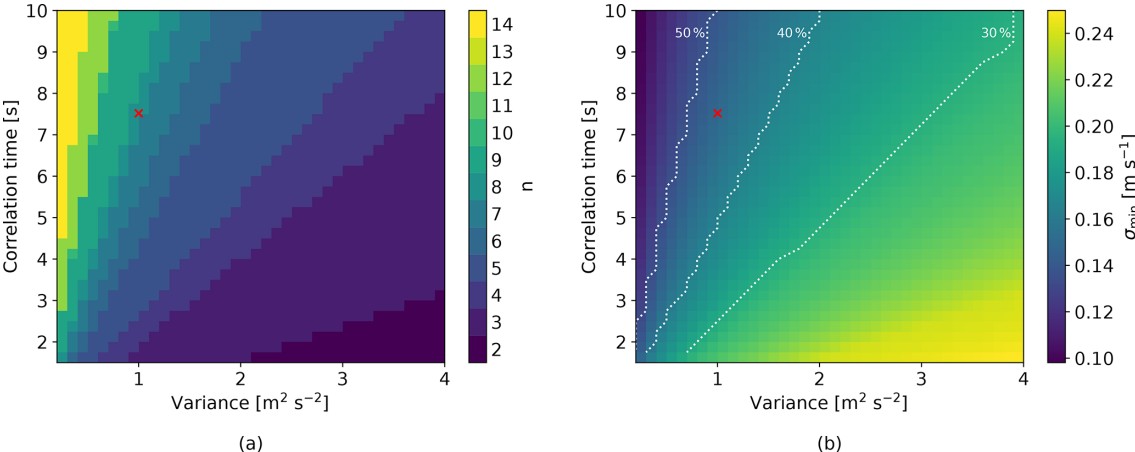

**Figure 7. (a)** Length of the filter window $n$ for which the measurement uncertainty of the transverse wind components gets minimized by a post-process low-pass filtering, i.e., averaging. The minimum depends on the correlation time and variance of the synthetic data set used for the uncertainty analysis (input parameters are the same as for the other figures; see Table 2). **(b)** Value of the minimized measurement uncertainty for various values of correlation time and variance. The values of uncertainty, which are 30 %, 40 %, and 50 % less than the uncertainty without post-processing ($\sigma = 0.28\,\mathrm{m\,s^{-1}}$), are plotted with dashed white lines. The red cross indicates the default configuration (standard values of variance and correlation time; see Table 2) in both plots.

mental wind data from MPCK measurement campaigns as raw measurement data, it could be shown that for typical conditions in an MPCK measurement campaign, it should be possible to achieve measurement uncertainties of around $\sigma = 0.15\,\mathrm{m\,s^{-1}}$.

## 5.2 Comparison of results with experimental and synthetic wind data

For the comparison of the uncertainty propagation model with experimental and synthetic wind data, a value for the variance and correlation time of the turbulence was determined for each experimental data set. The method of characterizing the turbulence of experimental data is explained in Sect. 3.2. The turbulence characteristics were used as defining parameters for the generation of synthetic data, which allows comparing the results of the uncertainty analysis with experimental and synthetic data of similar turbulence characteristics. The analysis presented in this paper uses experimental data from an ultrasonic anemometer mounted either on the MPCK (Stevens et al., 2021) or on a ground-based measurement platform (Bertens, 2021). In the case of the ground-based wind velocity data, we expect a better prediction of the turbulence characteristics since the data include no oscillations due to movements of the measurement platform. Figure 8b shows the results of the uncertainty analysis with ground-based experimental wind data compared to the results based on a synthetic wind data set, generated with the turbulence characteristics of the experimental data set as an input parameter. The curves are very similar. This result validates the approach of using synthetic data sets for the uncertainty analysis and again shows the possibility of reducing the measurement uncertainty for the transverse components

in an actual experiment. Figure 8a shows the comparison between the results of the uncertainty analysis with MPCK data and the respective synthetic data set with the same turbulence characteristics. All curves show similar behavior for averaging lengths of up to five data points. For larger averaging lengths, the differences between the experimental data sets are in the same range as the differences between an experimental curve and its respective synthetic counterpart (same color). In the case of one data set (Flight 6), the curves deviate significantly.

## 6 Discussion

Due to the geometric constraints of the setup, the transverse components of the reconstructed wind vector initially suffer from rather high uncertainties. We discuss the mechanism of uncertainty reduction and how to decide on the best post-processing parameters in an actual experiment.

The results of the uncertainty analysis with synthetic wind data show that a reduction of the transverse uncertainty is possible when applying a post-processing low-pass filter (see Figs. 5 and 7). The minimum of uncertainty depends, besides some fixed system assumptions (see Table 2), mainly on the post-processing filter length and the characteristics of the measured wind fluctuations. The increase in averaging time has multiple effects on the measurement uncertainty. On the one hand, a longer averaging time can increase the uncertainty due to the loss of information about the dynamics of the data in the averaging time. On the other hand, it can decrease the uncertainty since every single measurement channel adds random statistically independent errors to the data. By averaging over multiple data points these statistically in-

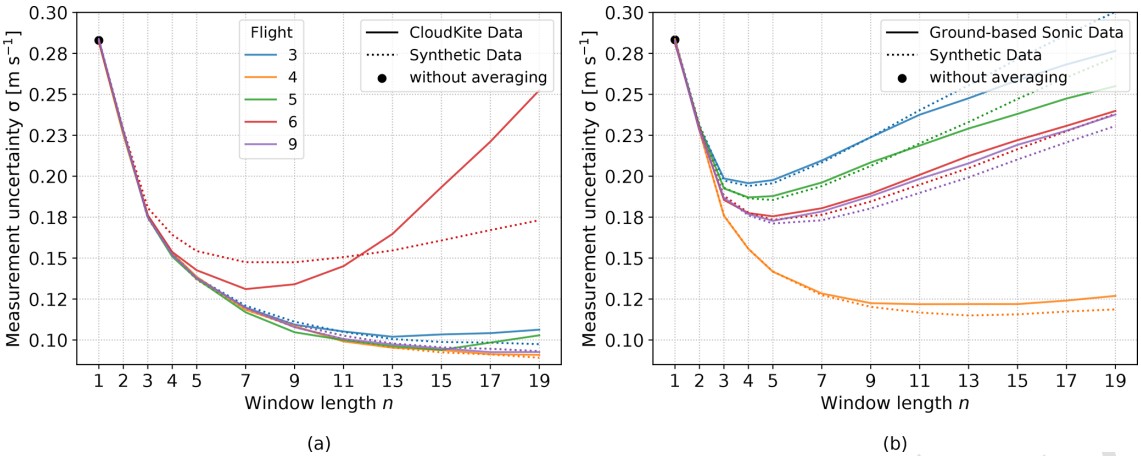

(a)  (b)

**Figure 8.** The uncertainty propagation model gives similar results when either synthetic data or experimental data are used. The plots show the measurement uncertainty of the transverse component ($x$) of experimental (bold lines) and synthetic wind data (dotted lines) with similar turbulence characteristics for varying post-processing window lengths. **(a)** Experimental wind data from an ultrasonic anemometer mounted on the MPCK, as well as synthetic wind data with similar turbulence characteristics, are used as input data for the uncertainty propagation model. The synthetic wind data are generated based on the characterization of the experimental data. Panel **(b)** shows the comparison between data from a ground-based ultrasonic anemometer (Bertens, 2021) and respective synthetic data sets with similar turbulence characteristics. The figure shows good consistency of the results of both data sets. Both figures use assumptions on the measurement system geometry and the measurement uncertainty as summarized in Table 2.

dependent errors can be reduced to some extent. If the filter window is too small the errors introduced by the measurement instrument do not average out. If the filter window is too large, wind fluctuations get smoothed out and we lose information; see Fig. 9. For small timescales, the fluctuation of the data is small compared to the uncertainty. Here, an uncertainty reduction is possible since averaging mainly impacts the errors and not the data. For longer averaging times, the averaging smooths wind fluctuations and the uncertainty increases again.

In Sect. 4.5 it could be shown that the ability to reduce uncertainties also depends on the measurement rate. If the measurement rate can be increased due to high aerosol density in the air the minimum of uncertainty achievable by post-processing averaging can be even further reduced. The random errors that define the intrinsic measurement uncertainty are added to each data point. The error fluctuation is therefore on the timescale of the measurement rate. On the other hand, the timescale of the wind fluctuation does not change for different measurement times. At higher measurement rates, it is therefore possible to average over more data points before smoothing the wind fluctuations of interest.

In an actual experiment, the system parameters like measurement rate, geometry, and measurement uncertainty are known and/or predicted based on profound knowledge. However, for determining the post-processing parameters to minimize the measurement uncertainty, knowledge of the timescales and size of fluctuations of the measured wind data, i.e., the turbulence characteristics, is additionally required.

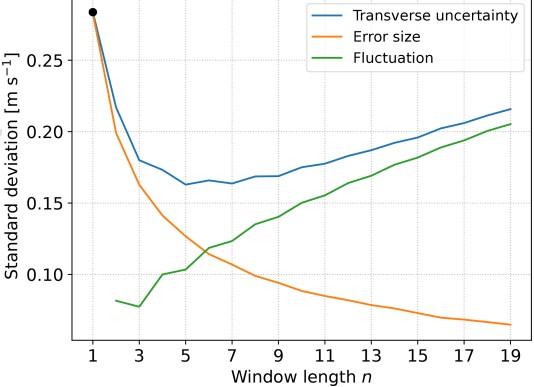

**Figure 9.** The mechanism behind the uncertainty reduction is based on the interplay between the timescales and magnitudes of measurement errors and wind fluctuations. A synthetic wind data set in the default configuration (see Table 2) is used and post-processed with a simple moving average (box filter) with varying window lengths, i.e., timescales. The analysis is done for the transverse $x$ component. The blue curve ("Transverse uncertainty") shows the measurement uncertainties of this data set for different window lengths, calculated with the uncertainty propagation model. The orange curve ("Error size") shows the influence of the averaging on the errors introduced by the lidar measurement. For this, the standard deviation $\sigma$ of the distribution of the error size is plotted when applying a simple moving average with varying window length $n$. The green curve ("Fluctuation") is calculated as follows: each data point of the initial wind data is subtracted from the mean around this data point, calculated with a given window length $n$. The standard deviation of these values is a measure of the fluctuations.

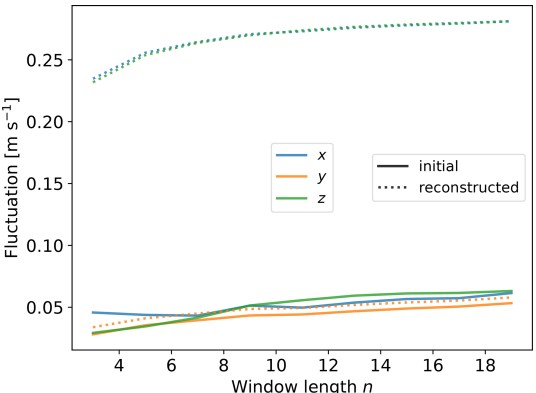

**Figure 10.** The longitudinal component ($y$) of the measured wind data can be used for determining optimal post-processing parameters. This figure shows the fluctuation of an initial wind data set ("initial", bold lines) and the fluctuation of the data set reconstructed from the "erroneous" measurement data ("reconstructed", dotted lines) for all three spatial components. The fluctuations are defined as the standard deviation of the distribution of the deviations of a data point from the mean of several data points. The $x$ axis denotes the averaging length for calculating the mean. The reconstructed data set is calculated as explained in Sect. 3.3. The initial data set is the experimental MPCK data set "Flight 3" (3D sonic anemometer; see Sect. 3.2).

The challenge in finding the turbulence characteristics of the measured wind to determine post-processing parameters is that the contribution of the statistical errors mostly dominates the data fluctuations for the transverse components ($x, z$), which is illustrated in Fig. 10. Here, the sizes of the fluctuations of an initial data set and a reconstructed data set are shown. The latter suffers from errors introduced by the measurement, which are geometrically "amplified" for the transverse components ($x$ and $z$). Thus, the turbulence characteristics of the transverse components cannot be directly derived by analyzing the reconstructed data set. However, this is possible with sufficient accuracy for the longitudinal component. We could therefore use the turbulence characteristics of the longitudinal component ($y$ component) to determine the best $n$ (post-processing filter length) for all components if we assume that the fluctuations in all three spatial components are similar. We could then run the uncertainty analysis with a synthetic data set defined by the assumed turbulence characteristics. The best $n$ can then be found by determining the minimum in uncertainty like in Fig. 5 or by determining the intersection of the fluctuation of the wind data with the curve of the error fluctuation, as plotted in Fig. 9. It needs to be validated whether the approach of using the turbulence characteristics of the longitudinal component to determine the post-processing parameters of the transverse components is generally valid, i.e., for all data sets and typical atmospheric conditions. This will be addressed in upcoming measurement campaigns with the CTL.

In Sect. 5 we present the results of the uncertainty analysis with experimental data and conclude that the accuracy of the characterization is mostly sufficient for a correct prediction of the reduction of uncertainty due to post-processing. However, the determination of the correlation time, especially in the case of airborne wind data, should be interpreted with caution. The correlation time is determined based on the autocorrelation function of the wind data, calculated as explained in Sect. 3.2. We saw that in the characterization procedure, the autocorrelation function does not completely decay to zero, even when using the entire data set as input values. The function still oscillates around zero, even for long lag times. Thus, a value for the correlation time can only be vaguely estimated. For a more precise determination of the correlation time, a profound post-processing is needed to filter out oscillations from the measurement platform and choose segments for which the correlation time can be regarded as constant. Nevertheless, it could be shown that the quality of the determination of the correlation time is mostly sufficient for a good match of results between experimental and synthetic data, especially when using wind data from a ground-based sonic anemometer, which does not suffer from platform-induced oscillations.

Another approach to determine the best $n$, i.e., the best post-processing filter length, is also possible without the characterization of the measured data. The theoretical analysis of the measurement uncertainties (Sect. 4) shows which $n$ leads to an uncertainty reduction for typical atmospheric conditions. In Fig. 7 we show that for a wide range of turbulence characteristics, the uncertainty of the transverse component can be reduced. The minimum is reached for the most data sets when using a filter length of $n = 3$–$9$. Also, the experimental results (Sect. 5) show that a post-processing with a filter length in this range reduces the uncertainty of all experimental data used. Thus, choosing a post-processing filter length of $n = 5$ is a reasonable choice for the CTL or similar multi-beam lidars with the assumptions defined in Table 2. Figure 11 shows that the transverse uncertainty is reduced by at least 30 % for all experimental data given. In an experiment with unknown wind characteristics a reduction of 30 %–50 % can thus be expected. The uncertainty of the longitudinal component will always increase but stays below the uncertainty of the transverse component. It is also possible to apply the post-processing only to the transverse component if differentiated data processing of the components does not cause problems for further use.

## 7 Conclusions

In this work, the measurement uncertainty of the CTL or similar multi-beam wind lidar systems was analyzed. The CTL has three optical heads which are spatially separated and focused on one point in a defined distance ($< 50$ m). The lidar is designed for mounting on airborne platforms

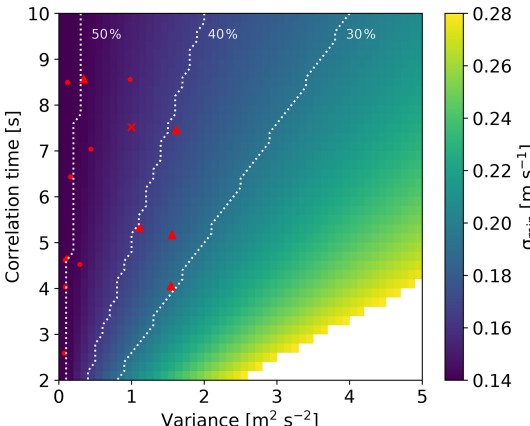

**Figure 11.** Uncertainty of transverse component ($x$) by applying a post-processing Gaussian filter with a window length of five data points ($n = 5$). The uncertainty is reduced for nearly all values of variance and correlation time used. White indicates uncertainties larger than the uncertainty without averaging ($0.28\,\mathrm{m\,s^{-1}}$). The red cross indicates the turbulence characteristics of the synthetic data set used for all plots in this paper (default configuration; see Table 2), unless otherwise stated. The red dots indicate the results of the characterization of the MPCK field campaign wind data. The red triangles indicate the characterization of the ground-based sonic anemometer wind data sets. The white lines show the values at which the uncertainty is reduced by 30 %, 40 %, and 50 % of the uncertainty without post-processing ($\sigma = 0.28\,\mathrm{m\,s^{-1}}$).

like the MPCK. To derive a 3D wind vector from the data of the three spatially separated lidar telescopes a reconstruction algorithm is needed, which is presented in this paper. An uncertainty propagation model is introduced which reveals the dependencies of the measurement uncertainty on system design and wind characteristics. The model was tested with synthetic wind data generated based on an Ornstein–Uhlenbeck process, as well as with experimental wind data from an MPCK measurement campaign and from a ground-based sonic anemometer. The spatial components of the reconstructed 3D wind vector in the transverse directions ($x, z$) to the main lidar direction have a high uncertainty due to the geometric amplification of measurement-introduced statistical errors in the reconstruction process.

A post-processing approach was introduced that consists of applying a Gaussian low-pass filter to reduce the statistically independent errors of the individual measurement channels, which can be considered averaging over multiple data points. This post-processing filters out statistically independent errors but at the same time smoothes out wind fluctuations on a certain timescale. Nevertheless, the uncertainty of the 3D wind measurement can be reduced for typical wind conditions (correlation time values ranging from 1–10 s and variance values of 0–5 $\mathrm{m^2\,s^{-2}}$) and for the assumptions on the system design (measurement rate, measurement uncertainty, etc.) and geometry (telescope separation and focus distance).

It could be shown that the characterization of the measured data to determine the best post-processing parameter can be challenging in an actual experiment. However, even without precise knowledge of the turbulence characteristics, it turned out that a reduction by around 30 %–50 % of the measurement uncertainty of the transverse wind component can be expected when averaging over five data points. The resulting measurement uncertainties for the CTL are $< 0.2\,\mathrm{m\,s^{-1}}$ for all spatial components. These results are valid for a multi-beam wind lidar with parameters comparable to the CTL (telescope separation, focus distance, measurement rate, measurement accuracy, etc.) as well as for a wide range of turbulence characteristics and thus for typical wind conditions.

Highly resolved 3D wind measurements with the Cloud-Kite Turbulence LiDAR or other multi-beam, airborne-mounted wind lidars are thus possible and useful for turbulence research.

## Appendix A: Consideration of other sources of measurement uncertainty

In this section, we present the estimation of potential error sources other than the measurement error that we focus on in the main text.

### A1 Geometric tolerances

We expect this to be a negligible source of error since the precise geometric dimensions of the measurement frame can be measured before mounting of the device to the CloudKite balloon. This includes the distances between the telescopes (side length) and also the distance and lateral position of the foci, which are straightforward to measure in a laboratory setting with millimeter accuracy. The analysis presented here also assumes that all three beams hit the focal volume under the same angles, which is more intricate to ensure. A geodetic instrument like an absolute tracker can be used to precisely measure all coordinates (instrument and foci) in 3D space with an accuracy far better than 1 mm. From this calibration procedure the angles can be extracted and compensated for.

### A2 Influence of wind on the measurement geometry

The spatial resolution, i.e., the measurement volume, is assumed to be 1 $\mathrm{m^3}$. This results from the foci being significantly longer (about 1 m) than their lateral dimensions. During alignment of the setup, before mounting, all three foci are superimposed onto one point through the use of deflection mirrors in the telescope heads.

The change in angular orientation of one single telescope required for its focus to move by 0.5 m, i.e., half the spatial resolution, can be estimated. As illustrated in Fig. A1, the change in angular orientation can be approximated by $\gamma = \arctan\left(\frac{\Delta x}{d_\mathrm{f}}\right)$, where $\Delta x$ is the change in lateral position of the

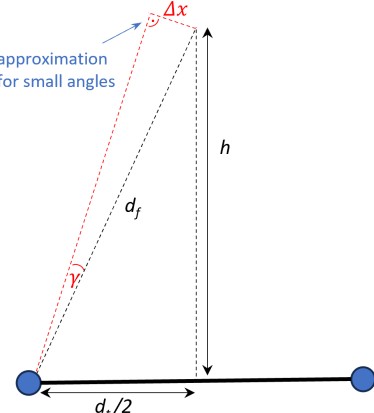

**Figure A1.** 2D schematic for the estimation of the change in lateral focus position ($\Delta x$) depending on the change in angular orientation ($\gamma$) of one telescope head (blue circle).

focus and $d_f = 15\,\mathrm{m}$ is the distance of the focus. For $\Delta x = 0.5\,\mathrm{m}$, this yields $\gamma = 1.9°$. Considering the stiffness of two connected carbon tubes (see Fig. 1) and the very small attack surface for the wind, $1.9°$ seems like an unrealistically high value for bending due to wind, which is why we think this error is also of minor importance.

## A3    Influence of temperature on the measurement geometry

Concerning the effect of temperature, we assume operating temperatures between 0–40 °C and alignment of the setup under lab conditions at 20 °C. Thus, a maximal change in temperature of 20 °C must be considered. The temperature extension coefficient of carbon is $2 \times 10^{-6}\,\mathrm{K}^{-1}$. Considering the longest dimension, i.e., the 3 m bars between the telescopes, this results in a maximal change in length of merely 0.12 mm, which is negligible.

## A4    Dynamic tolerance due to platform motion

With dynamic tolerance we refer to the fact that the Cloud-Kite and the attached measurement device are moving during the actual measurement. There are several points to consider here: first, it should be mentioned that the absolute location (in world coordinates) of the point of measurement does not have to be known precisely for these types of measurement.

Second, the influence of the motion during the acquisition of a single data point, i.e., during 100 ms TS7, must be considered. It is known from previous measurement campaigns that the CloudKite platform motion has its main frequencies around 1 Hz (Schröder, 2023). This is 1 order of magnitude slower than the acquisition of a single data point. However, there might still be some movement within 100 ms TS8. This can be regarded as an increase in the actual measurement volume.

Third, there is platform motion during the whole measurement run, which might last up to many hours. This leads to a motion of the focus, i.e., the point of measurement. This motion can be tracked using inertial measurement units (IMUs). For this reason, two IMUs in each telescope head are integrated into the measurement device. Whether this also allows for the correction of the tracked movement depends on the parameter of interest in the post-processing. For example, the mean wind velocity could be corrected for the platform motion. For other parameters it can be more intricate or even impossible. However, this is an error source that influences the analysis of the measured data but hardly the individual measurement data points. Therefore, a detailed analysis of the consequences of this platform motion is beyond the scope of this paper.

## Appendix B: Measurements with a high signal-to-noise ratio

To investigate the distribution of velocity measurements, experiments under defined laboratory conditions were performed. Data from the internal reference channel were analyzed. Also, measurements with a single telescope, i.e., one measurement channel, on a hard target (laboratory wall) placed in the focus were done. Figure B1 shows the histograms of the peak positions of 1000 measurements for the reference channel (Fig. B1a) as well as for the hard target (Fig. B1b) and a Gaussian fit. Both histograms approximate a normal distribution.

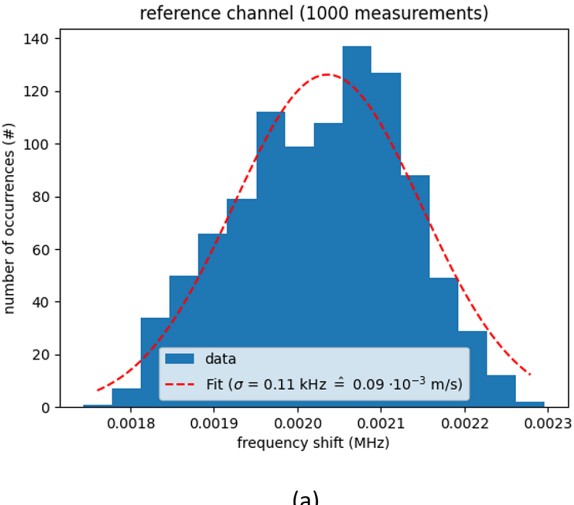

(a)

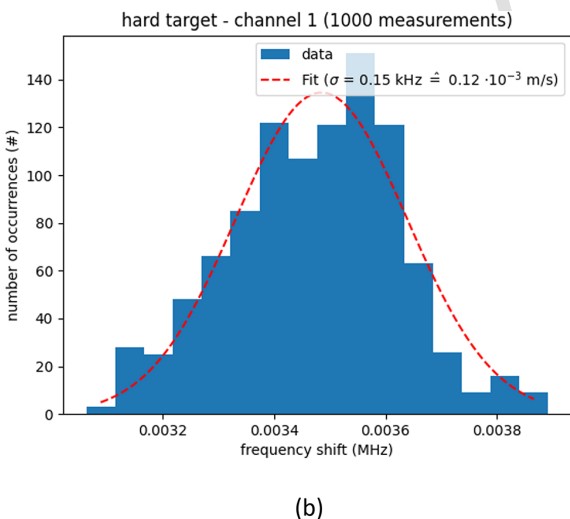

(b)

**Figure B1. (a)** Histogram of 1000 measurement values extracted from the reference channel. The measured frequency shifts, which correspond to a velocity, approximate a normal distribution with standard deviation $\sigma_f = 0.11$ kHz. This equals $\sigma_v = \lambda \sigma_f/2 = 0.09 \times 10^{-3}$ m s$^{-1}$, where $\lambda = 1545$ nm is the laser wavelength. **(b)** Histogram of 1000 measurements performed on a hard target (lab wall) placed in the focus of measurement channel 1. The measured frequency shifts approximate a normal distribution with standard deviation $\sigma_f = 0.15$ kHz. This equals $\sigma_v = \lambda \sigma_f/2 = 0.12 \times 10^{-3}$ m s$^{-1}$.

## Appendix C: Simulation of the effect of white noise on the peak position

To investigate the influence of a low signal-to-noise ratio (SNR) on the fluctuations of the peak position in the measured frequency spectra, a numerical Monte Carlo simulation was performed. Data sets of 4096 samples were generated with a beat frequency of 5 MHz. White noise was added to simulate the shot-noise-limited measurement regime. The data were then multiplied with a flat-top window and then fast-Fourier-transformed. The absolute values of 4000 spectra were averaged to get one target spectrum. From this target spectrum the peak position was extracted using a Gaussian fit. Also, the SNR was calculated by dividing the peak height by the standard deviation of the background noise. Figure C1a shows the distribution of the obtained frequency shifts for an SNR of 6.1, as might be the case in actual wind measurements. A Gaussian fit yields a standard deviation that corresponds to $\sigma_v = 0.009\,\mathrm{m\,s^{-1}}$. Figure C1b shows that the standard deviation decreases with increasing SNR.

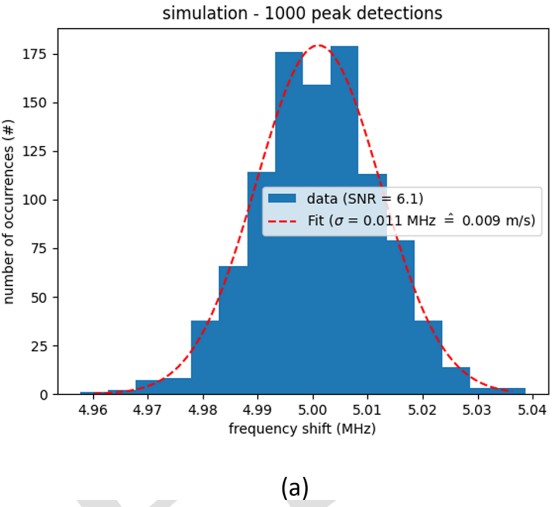

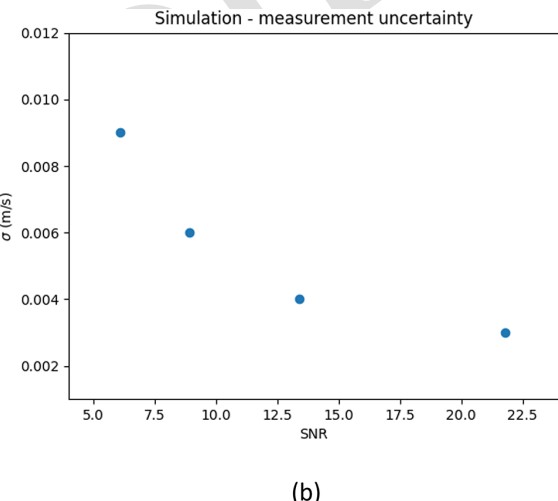

(a)                                         (b)

**Figure C1. (a)** Histogram of 1000 simulated measurements with white noise and an SNR of 6.1. **(b)** Measurement uncertainty as obtained from the fitted histograms for simulations with varying SNR.

## Appendix D: Comparison of averaging implementations

Figure D1 shows the comparison of four low-pass-filter implementations, which correspond to an averaging over data points. The data set is convolved with a certain filter function. The function can either be a window function of a given length with defined weights (uniform and triangular) or a function with a given shape, e.g., Gaussian. Here we compare a uniform window, which corresponds to a simple moving averaging, a triangular window, a Gaussian filter, and a so-called Butterworth filter. The Butterworth filter is implemented as a low-pass filter in frequency space and applied forwards and backwards to reduce phase delays and have a pass band as flat as possible. The Gaussian filter is defined as explained in Sect. 3.4 with a standard deviation of $\sigma^{\mathrm{filt}} = \frac{n}{4}$ and truncated to a window length of $n$.

One can observe in Fig. D1 that, in the case of filtering with a window function, the uncertainty is lower for an odd number of averaging lengths than for even numbers. Comparing the processed data set with the initial data set requires assigning the data points to each other point by point. The result of an averaging over a segment has to be assigned to the data point in the middle of this segment. In the case of even numbers, the resulting data set is shifted compared to the initial data set due to the abundance of an index in the middle of the averaging segment. This behavior introduces an additional error. This error has no physical origin, but for better interpretation of the results, the Gaussian filter is used for the results in the present analysis, which does not suffer from these errors.

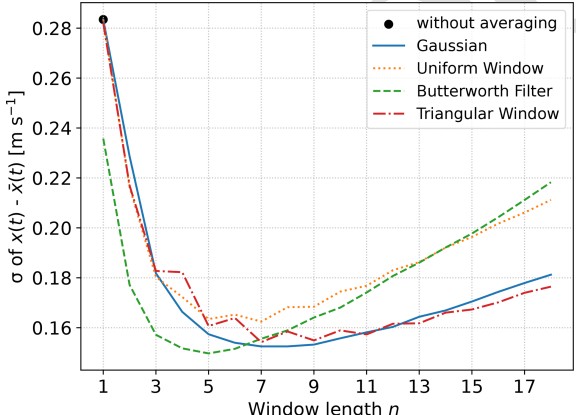

**Figure D1.** The figure shows the comparison of different post-processing implementations. A synthetic input data set is used as the input data set for the uncertainty propagation model explained in Sect. 3.3. The results with different post-processing algorithms, i.e., different filtering implementations, are plotted. Only the transverse component ($x$) of the reconstructed 3D wind vector is shown. The black dot indicates the measurement uncertainty without averaging.

## Appendix E: Optimizing mounting geometry

One goal of this work is to optimize the data quality of an airborne-mountable 3D wind lidar. To this end, it is also worth investigating the geometrical configuration which yields the highest measurement accuracy. The uncertainty depends on the angle between the line-of-sight directions of the three telescopes and the spatial component of interest. This angle is determined by the telescope distance $d_{\mathrm{t}}$ and the focus distance $d_{\mathrm{f}}$ (see Sect. 2.3). Figure E1 shows the transverse measurement uncertainty in terms of the standard deviation depending on the focus distance and the telescope distance, respectively. One can observe a nearly linear relation between the focus distance and the measurement uncertainty of a reconstructed spatial wind speed component. The dependence of the measurement uncertainty on the telescope distance follows an reciprocal decay. To optimize the measurement uncertainty, the telescope distance should be maximized and the focus distance should be minimized within the given limitations. Furthermore, the result shows that due to the reciprocal behavior of the uncertainty dependency on the telescope distance, the value of 3 m for the telescope distance is a good compromise since for larger distances the uncertainty decreases only slowly (second decimal place).

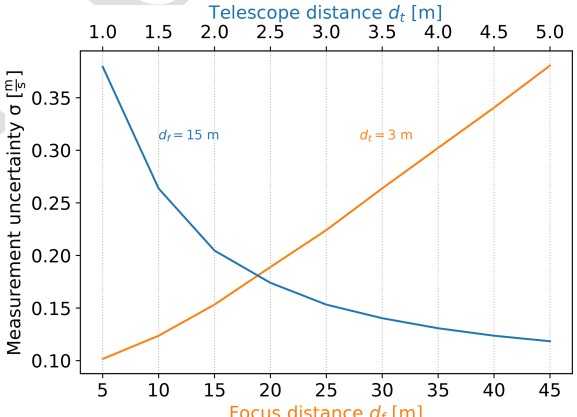

**Figure E1.** The measurement uncertainty of the transverse component ($x$ component) of the reconstructed wind vector as a function of geometrical parameters of the CTL. The telescope distance is set to 3 m for changes in the focus distance (orange). The focus distance is set to 15 m for changes in the telescope distance (blue). The same input data and parameters were used as defined in Sect. 4.1. The results of this plot are based on the uncertainty propagation model based on synthetic wind data and include a post-processing Gaussian filter with a window length of six data points ($n = 6$).

*Code and data availability.* The experimental data as well as the numerical Python code used for the simulations can be shared upon legitimate request.

*Author contributions.* GB, PvO, and MW initialized the idea for the presented analysis. MW conceptualized the analytical approach. WK developed the model code and performed the formal analysis. WK prepared the manuscript with contributions and reviews from all co-authors. PvO, GB, and MW provided supervision, validation, and project administration.

*Competing interests.* The contact author has declared that none of the authors has any competing interests.

ther geographical representation in this paper. While Copernicus Publications makes every effort to include appropriate place names, the final responsibility lies with the authors.

*Acknowledgements.* We thank the other members of the TWISTER team for initializing and/or continuously supporting the project: Tobias Bätge, Eberhard Bodenschatz, Karsten Buse, Björn Klaas, Venecia Chávez Medina, Katharina Predehl, Alexander Reiterer, Oliver Schlenczek, and Marcel Schröder. We also thank Marcel Schröder and Augustinus Bertens for providing experimental wind data. We thank Marcel Schröder for his work preparing the data for our analysis. We thank Valentin-Vierhub Lorenz and Christoph Werner for fruitful discussions on the measurement uncertainty of the instrument.

*Financial support.* This research has been supported by the Fraunhofer and Max Planck cooperation programme (project TWISTER – turbulent weather in structured terrain).

*Review statement.* This paper was edited by Gerd Baumgarten and reviewed by three anonymous referees.

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

**Remarks from the language copy-editor**

CE1    Please confirm the change. Our standard is to include metric equivalents of imperial units.

**Remarks from the typesetter**

TS1    Thank you for providing the emails. Please note, however, that changes like these need to be approved in a "Post-review adjustments" process to retain transparency for the reader. Therefore, we kindly ask you to summarize all relevant changes in one file which will be forwarded to the editor. Please note that the status of your paper will be changed to "Post-review adjustments" until the editor has made their decision. We will keep you informed via email.

TS2    Due to the requested changes, we have to forward your requests to the handling editor for approval. To explain the corrections needed to the editor, please send me the reason why these corrections are necessary. Please note that the status of your paper will be changed to "Post-review adjustments" until the editor has made their decision. We will keep you informed via email.

TS3    Please see previous remark regarding editor approval.

TS4    Regarding the addition of "$^a$", we have to forward your requests to the handling editor for approval. To explain the corrections needed to the editor, please send me the reason why these corrections are necessary. Please note that the status of your paper will be changed to "Post-review adjustments" until the editor has made their decision. We will keep you informed via email.

TS5    As explained above, please note that changes like these need to be approved in a "Post-review adjustments" process to retain transparency for the reader. Therefore, we kindly ask you to summarize all relevant changes in one file which will be forwarded to the editor. Please note that the status of your paper will be changed to "Post-review adjustments" until the editor has made their decision. We will keep you informed via email.

TS6    For the new versions of Figs. 3 and 4, we have to forward your requests to the handling editor for approval. To explain the corrections needed to the editor, please send me the reason why these corrections are necessary. Please note that the status of your paper will be changed to "Post-review adjustments" until the editor has made their decision. We will keep you informed via email.

TS7    Please clarify if meters per second ($m\ s^{-1}$) or milliseconds (ms) is meant here.

TS8    Please clarify if meters per second ($m\ s^{-1}$) or milliseconds (ms) is meant here.