# Peer review of "Analysis of the measurement uncertainty for a 3D wind-LiDAR"

_Atmospheric Measurement Techniques, 2023_

## Referee Comment (RC2)

Referee Report on
**Analysis of the measurement uncertainty for a 3D wind-LiDAR**

The article presents an error propagation analysis for the Cloud Turbulence Lidar (CTL). The analysis uses both synthetic and measured time series to assess the variability that could be expected in the measured time series due to the uncertainty in the 3 detectors reported longitudinal velocity. This error is treated as Gaussian and added to the longitudinal velocity which should be measured by each detector for the synthetic/experimental velocity. The 3 longitudinal velocities with added noise are then reconstructed to reproduce an approximation of the wind vector time series which would be measured, which is then compared to the original synthetic or experimental velocity.

The authors then use these approximated time series to test a filtering approach that can be used to post-process the added noise and reduce the random noise that has been artificially added in.

In general, the article is well written, and to the extent that the authors analysis is conducted, appears to be appropriate. However, there are several concerns that I regarding the applicability of this analysis to the general uncertainty output from the system. The biggest concern is simply that the uncertainty analysis only propagates a single error (specifically the detector error). As such, the study is very limited in its approach. Their own error propagation analysis provided in Section 3.6 indicates that the error is going to be amplified the precision of the support structure and resulting angle of the detectors to the plane of the lidar. The geometric dimensions of the array are assumed to be precisely known and constant. What is the sensitivity of the wind to uncertainty in the dimensions? Can this structure be assumed to be perfectly rigid during a measurement? Does the measuring distance, $h$, impact uncertainty? Surely temperature changes will result in some expansion/contraction of the support structure? Furthermore, uncertainty in the Euler angles used to transform the wind velocity from the lidar frame of reference to the inertial frame of reference can produce significant uncertainty in the resulting wind components. This effect is also not considered.

Therefore, as noted, this manuscript is somewhat limited to simply the propagation of the detector uncertainty to the measured wind in the lidar plane of reference, with an analysis of the filter/smoothing functions best suited to reduce this added noise. In this context, careful characterisation of the detector uncertainty would be important for the analysis. However, this characterisation is limited to just two sentences, which does not sufficiently justify the stated detector uncertainty of $\sigma^{det} = 0.04 \text{ m s}^{-1}$. From what I can tell, the authors take the resolution of the sensor output of $0.1 \text{ m s}^{-1}$ and assume a 99% confidence bound(?) of $3\sigma$ to get a standard deviation of the uncertainty of $0.04 \text{ m s}^{-1}$ (assuming that they are rounding up?). However, this is just guesswork on my part. If this is the case, than the error propagation analysis is not even assessing the uncertainty of the individual lidar measurements, but is simply assessing the propagation of the resolution limitations of the individual lidar measurement.

I would therefore recommend that the authors, at the very least, provide more detail and care into the assessment and description of $\sigma^{det}$. The paper would also be much more strengthened by including additional error sources into their analysis, however this may require significant revisions of the manuscript.

Additional comments:

1. Figure 2 was a little confusing for me due to the perspective. Specifically it took me some time to understand that the plane of the lidar was parallel to the oncoming wind field. I think the confusion comes from the kite being angled to the mean wind, but the lidar appearing to be drawn on the kite. Once I had figured out the arrangement of the CTL on the MPCK, the text of section 2.2 made more sense, but perhaps the authors may wish to add more details/different views to Figure 2 so that others may not be equally confused.

2. Tables 1 and 2 have redundant information and could be combined. Note that whereas Table 2 refers to 10 Hz as the sampling rate, Table 1 refers to the same quantity as the time resolution. Technically, the time resolution is 0.1 s, not 10 Hz.

3. line 182: should be 'lose', not 'loose'.

4. The synthetic time series input the noise as a white noise process, what justification is there that the detector uncertainty appears in the form of white noise. Note that the nature of the noise could impact the efficacy of the smoothing for noise removal/uncertainty reduction.

---

## Author Comment (AC1)

**Analysis of the measurement uncertainty for a 3D wind-LiDAR - Reply to RC1 from Referee #1**

Wolf Knöller[1], Gholamhossein Bagheri[2], Philipp von Olshausen[1], and Michael Wilczek[2,3]

[1]Fraunhofer Institute for Physical Measurement Techniques IPM, Freiburg, Germany
[2]Max Planck Institute for Dynamics and Self-Organization (MPI-DS), Göttingen, Germany
[3]Theoretical Physics I, University of Bayreuth, Bayreuth, Germany

August 30, 2024

**Review:** *The authors present an analysis of how Normally distributed errors in line-of-sight velocity measurements and turbulence can potentially impact the error in 3D wind estimates for the CloudKite Turbulence LiDAR (CTL). The concept proposed for this work is very important to building wind sensors and is a necessary analysis to such an instrument development. It's relevant in scope for AMT if the analysis is conducted accurately and robustly. However, I am not convinced that the simulated errors are modeled correctly. As such, the analysis becomes circular where the assumed noise characteristics are artificially imposed on both the simulated and experimental data and the efficacy of the solutions are reliant on the assumed model being accurate.*

*The central issue with this manuscript is how the wind uncertainty is modeled. I am skeptical that it represents an accurate description of uncertainty in such a wind measurement system. If the authors are correct in how the noise is modeled, the manuscript needs to include a robust justification. If this noise model is incorrect, the analysis needs to be updated so that it accurately models the instrument and its uncertainties.*

*What follows are four points regarding the content of the manuscript (with one aside on the instrument). The most important point, and most significant barrier to my approving this for publication is the first item. Most of the other topics are tangentially related to my concern with the noise model.*

**Reply:** Thank you for raising these points while confirming the importance of modeling such a system. We address your concerns in the following sections.

**Review:**

***Noise model***

*Stated in section 2.1: "As the detection is shot noise limited, a Gaussian distribution is assumed, for which $\sigma = \text{FWHM}/(2\sqrt{2ln2})$. Consequently, a detector uncertainty in terms of standard deviation can conservatively be estimated to be $\sigma_{det} = 0.04$ ms$^{-1}$. Also, the LiDAR system has an internal reference channel which suggests that the detector noise is even below $0.04$ ms$^{-1}$"*

*Detectors don't output velocities. Detector noise manifests as an analog voltage at the output of the detector. That output is typically digitized, and stored as a fixed length time series, passed through an FFT operation to which a peak finding algorithm is applied. The corresponding peak frequency corresponds to a line-of-sight velocity dependent on the wavelength of the light. I cannot intuitively see how Poisson noise (shot noise is Poisson not normally distributed) results in Normally distributed independent line-of-sight velocities. Furthermore, I would expect uncertainty in velocity is highly dependent on the amount of aerosol loading in the scattering volume. Typically peaks in spectra resulting from air motion are very easy to find when the backscatter is high (significantly above variations in the noise floor and those systematic variations in the baseline we all pretend don't exist in our instruments), but as the peak height drops near the noise floor, the velocity error tends to increase rather rapidly (and is probably not normally distributed). In order to increase the signal to noise ratio, one typically does not average velocity estimates (as done in this analysis), but instead averages FFT spectra before applying the peak finding algorithm. Factors that would tend more towards the uncertainty described in this manuscript would have to do with the resolution of the system (which means subsequent samples are not generally independent and therefore effectively suppressed with averaging). Those would be terms such as laser line width, windowing function and the FFT sequence length.*

*There is a caveat to my comments here because while I can say that I have analyzed 3D wind sensor data before (not unlike what is described here) that instrument did not employ FMCW. It is not entirely clear to me if I am missing how FMCW would alter the way noise manifests in the eventual line-of-sight velocity estimate. If FMCW alters this, such that the presented noise model is a valid representation of the effects of shot noise, the authors need to robustly show it. As part of this, it would be very helpful if they would provide a more thorough description of the FMCW scheme being employed (e.g. specific modulation scheme) and how the raw data is processed to obtain a line-of-sight wind measurement.*

**Reply:** We appreciate the criticism as more precision in our assumption and derivations is needed. For this reason, we've added a new section, 2.2 Measurement uncertainty, to the manuscript which explains the assumptions in detail.

The first issue is our assumption of normally distributed line-of-sight velocity measurements. We agree with the reviewer's description of what happens with the detector voltage to ultimately yield a velocity. No FMCW specialties come into play here. On the detector we have the coherent interference of the local oscillator with the signal. The local oscillator is adjusted so that it is the dominant noise term, hence we are in the shot noise limited detection scheme. Consequently, there are many photons arriving on the detector (more precisely: on each of the two balanced photo detectors). Also, having a peak coming out of the noise floor (otherwise the wind velocity can not be extracted, which can happen in reality, of course) means that there is also a significant number of signal photons involved. For large numbers of photons the Poisson distribution approximates the normal distribution very well.

Now the subsequent question is: Does the dominance of white noise lead to an uncertainty of the peak position that is normally distributed? We have performed a simple simulation of a beat signal with added white noise, averaged 4000 spectra as in a typical measurement, and analysed the peak positions which were found by fitting a Gaussian function (see Appendix C). The result shows two things:

1. The peak positions and thus the velocities follow a normal distribution. It makes intuitively sense that the uncertainty of the peak position follows a symmetric function. The Poisson distribution is not symmetric for small mean values, but it approximates the normal distribution well for large mean values.

2. Increasing the level of added white noise, i. e. decreasing the signal-to-noise (SNR) ratio, leads to an increase in the fluctuation of the peak positions (the measured wind velocity). This confirms the reviewer's assumption that a decreasing SNR leads to increased uncertainty. However, it remains normally distributed for low SNR.

Regarding the three potential noise sources that are mentioned:

▷ Laser line width: We assume a typical measurement distance of $d = 15$ m. The resulting travelling time of the photons is $t = 2 \cdot d/c \approx 100$ ns. On this timescale, the laser is very stable.

▷ Windowing function: Every slice of raw data is multiplied with a Flattop window function before performing the FFT. We have a sampling rate of 491.52 MHz and typical chunk size of 4096 data points which yields a frequency spacing of 120 kHz. As the main lobe of a Flattop window in frequency space is about 4 bins wide (full width half maximum), we get a width of the peak (one $\sigma$) of approximately $2 \cdot 120\text{kHz} = 0.24\text{MHz}$. This broadens the peak but does not influence its position.

▷ FFT length: As mentioned in the previous paragraph, a typical FFT length is 4096 data points. This results in a frequency spacing of $df = 120$ kHz which equals a velocity uncertainty of $dv = \lambda \cdot df/2 = 0.093$ ms$^{-1}$, where the wavelength is $\lambda = 1545$ nm. This provides an order of magnitude for the velocity uncertainty and ignores the possibilities of sub-sampling resolution due to peak fitting.

A FMCW scheme is different in detail from a CW or a pulsed wind LiDAR but does not alter the general noise considerations discussed here.

We have added a brief description of the used FMCW scheme and a more thorough description of our derivation of the measurement uncertainty to the manuscript.

**Review:** *Dependence on Sampling Rate*
*The section investigating a dependence on sampling rate does not strike me as an accurate analysis of the tradeoffs in sampling. Inherently moving to a higher sampling rate will*

> ▷ *Change the frequency resolution and Nyquist frequency of the FFT*

> ▷ *Change the noise characteristics*

*An analysis using the same noise model and assuming the same Normally distributed noise characteristics basically assumes that one gets something for free from higher sampling rates. That's not the case. In general, assuming the detector and anti-aliasing filters can support the increased bandwidth, one would expect the noise amplitude to go up because what comes with increased sampling rates is shorter integration times and comparatively less noise suppression.*

**Reply:** We thank the reviewer especially for this comment as it highlights a misunderstanding due to our poorly chosen wording. We are actually referring to a measurement rate, not a sampling rate. We have adapted this throughout the manuscript. Also, we have added a brief section on how the measurement value (wind velocity) is derived from the data.

When changing the measurement rate this means that a different number of individual spectra (derived by doing FFTs on sampled raw data sections) are averaged. The sampling rate, referring to the digitization of the voltage values of the detector, is not changed. Consequently, the frequency resolution and the Nyquist frequency are not changed when changing the measurement rate.

However, for a given signal the noise characteristics will generally change when changing the measurement rate. We assume that a different measurement rate is chosen due to different environment conditions (aerosol density). This means that the signal-to-noise ratio would be very comparable. We have clarified this in the manuscript.

**Review:** *Section 5.1 Uncertainty analysis with experimental wind data*
*The approach described in this section might make sense if my principle concerns about the analysis had to do with the accuracy of the wind simulations. However I'm concerned about the accuracy of the instrument noise model, which is not encapsulated in this analysis at all. To my perspective this is just another case of synthetic data, and does not reassure me that the authors are accurately modeling the instrument.*

**Reply:** Indeed, the analysis of experimental wind data is not done to validate the noise model. Rather, this analysis is included as an additional example, using experimentally measured wind values instead of simulated input. This also shows that the chosen approach holds true for the analysis of realistic wind data, here derived from an experimental campaign. As such, this underpins the approach of choosing an Ornstein-Uhlenbeck process to model wind.

**Review:** *Appendix A: Geometric tolerances*
*I was somewhat bothered by this statement: "We expect this to be a negligible source of error since the precise geometric dimensions of the measurement frame can be measured before mounting of the device to the Cloud-Kite balloon. This includes the distances between the telescopes (sidelength), but also the distance and lateral position of the focus." This isn't reassuring because it just says the authors assume it's not an issue. Since we all know there is no such thing as exact dimensions, it would be advisable to include an analysis of the tolerances in those dimensional measurements and how they propagate to uncertainty in the wind measurement. Some description of how the beam pointing is determined is also helpful (e.g. see 2.3.2 in Cooper et al 2016 – full reference below). Without that analysis, this just looks like a convenient assumption.*

**Reply:** While the three beams are easy to align to meet in one point in space in the lab, it will indeed be difficult to assure that all three beams are perfectly symmetrical and hit the common focal point under the exact same angle. Thank you also for the reference describing an approach to solve this problem.
Fraunhofer IPM has a 3D laser tracker available for exactly these types of measurement. It is regularly used for calibrating different types of multi-sensor systems. As a precise calibration provides all data to compensate for this error, we do not consider this effect essential for the analysis performed in this paper.
We have rephrased the paragraph in the appendix to be more precise.

**Review:** *Comment on the architecture*

*While not directly relevant to the review of this paper, I am concerned about the minimalist design of this instrument. Using 3 beams to measure a 3D wind vector leaves one relatively blind to unexpected error sources (this is based on personal experience). I would highly recommend that the CTL design include at least one additional beam. By having four line-of-sight velocity measurements, the estimation problem becomes over constrained and it creates a mechanism for detecting unexpected errors and biases in the instrument. This often becomes another source of irritation in the testing process (when you have to track down the errors you forgot to consider), but if you are committed to making an accurate measurement, it's essential insurance to catch those errors.*

**Reply:** Thank you very much for this advice and for sharing your experience. While having only three beams for measurement, we do have a fourth, internal channel (reference channel). This consists of a long glass fiber representing the signal path and a balanced photo detector where this signal and the local oscillator interfere. Of course, this does not over-constrain the measurement and is, consequently, not identical to having a fourth beam. However, it does provide some quality control of crucial system parameters, as e. g. the laser light source. This approach was chosen due to the constraints in available space and load (weight) on the Cloud Kite platform.

---

## Author Comment (AC2)

**Analysis of the measurement uncertainty for a 3D wind-LiDAR - Reply to RC2 from Referee #3**

Wolf Knöller[1], Gholamhossein Bagheri[2], Philipp von Olshausen[1], and Michael Wilczek[2,3]

[1]Fraunhofer Institute for Physical Measurement Techniques IPM, Freiburg, Germany
[2]Max Planck Institute for Dynamics and Self-Organization (MPI-DS), Göttingen, Germany
[3]Theoretical Physics I, University of Bayreuth, Bayreuth, Germany

August 30, 2024

As far as we can see, this review is identical with the one of round 1 of the discussions. We here provide our replies again for convencience and completeness.

**Review:** *The article presents an error propagation analysis for the Cloud Turbulence Lidar (CTL). The analysis uses both synthetic and measured time series to assess the variability that could be expected in the measured time series due to the uncertainty in the 3 detectors reported longitudinal velocity. This error is treated as Gaussian and added to the longitudinal velocity which should be measured by each detector for the synthetic/experimental velocity. The 3 longitudinal velocities with added noise are then reconstructed to reproduce an approximation of the wind vector time series which would be measured, which is then compared to the original synthetic or experimental velocity.*

*The authors then use these approximated time series to test a filtering approach that can be used to post-process the added noise and reduce the random noise that has been artificially added in.*

*In general, the article is well written, and to the extent that the authors analysis is conducted, appears to be appropriate. However, there are several concerns that I regarding the applicability of this analysis to the general uncertainty output from the system. The biggest concern is simply that the uncertainty analysis only propagates a single error (specifically the detector error). As such, the study is very limited in its approach. Their own error propagation analysis provided in Section 3.6 indicates that the error is going to be amplified the precision of the support structure and resulting angle of the detectors to the plane of the lidar. The geometric dimensions of the array are assumed to be precisely known and constant. What is the sensitivity of the wind to uncertainty in the dimensions? Can this structure be assumed to be perfectly rigid during a measurement? Does the measuring distance, h, impact uncertainty? Surely temperature changes will result in some expansion/contraction of the support structure? Furthermore, uncertainty in the Euler angles used to transform the wind velocity from the lidar frame of reference to the inertial frame of reference can produce significant uncertainty in the resulting wind components. This effect is also not considered.*

**Reply:** Thank you for raising this point. Several different error sources are mentioned by the reviewer that we have grouped into three sections that we address one by one in the following. We also adapted the manuscript to include these considerations (line 40f and appendix A).

**Geometric tolerances**

We expect this to be a negligible source of error since the precise geometric dimensions of the measurement frame can be measured before mounting of the device to the CloudKite balloon. This includes the distances between the telescopes (side length), and also the distance and lateral position of the foci, which are straightforward to measure in a laboratory setting with millimeter accuracy. The analysis presented here also assumes that all three beams hit the focal volume under the same angles, which is more intricate to ensure. A geodetic instrument like 3D laser tracker can be used to precisely measure all coordinates (instrument and foci) in 3D space with an accuracy even far better than 1 mm. From this calibration procedure the angles can be extracted and compensated for.

**Influence of wind and temperature on the geometry**

The spatial resolution, i. e. the measurement volume, is assumed to be 1 $m^3$. During alignment of the setup,

[Figure]

Figure 1: 2D estimation of the change in angular orientation of one telescope head required for a lateral change of its focus by 0.5 m.

before mounting, all three foci are superimposed onto one point by the use of deflection mirrors in the telescope heads.

The change in angular orientation of one single telescope required for its focus to move by 0.5 m, i. e. half the spatial resolution, can be estimated. As illustrated in figure 1, the estimated (2D approximation) change in angular orientation is 1.9°. Considering the stiffness of two connected carbon tubes (see Figure 1 in the main manuscript) and the very small attack surface for the wind, 1.9° seems like an unrealistically high value for bending due to wind which is why we think this error is also of minor importance.

Concerning the effect of temperature, we assume operating temperatures between 0 °C - 40 °C, and alignment of the setup under lab conditions at 20 °C. Thus, a maximal change in temperature of 20°C must be considered. The temperature extension coefficient of carbon is 2 $(10^{-6}\text{K}^{-1})$. Considering the longest dimension, i. e. the 3 m bars between the telescopes, this results in a maximal change in length of merely 0.12 mm, which is negligible, even considering the the lever of 15 m to the focus for changes in angular orientation.

**Dynamic tolerance**
With dynamic tolerance we refer to the fact that the CloudKite and the attached measurement device might be moving during the actual measurement. There are several points to consider here: First, we should mention that the absolute location (in world coordinates) of the point of measurement does not have to be known precisely for these types of measurement.

Second, the influence of the motion during the acquisition of a single data point, i. e. during 100 ms, must be considered. It is known from previous measurement campaigns that the CloudKite platform motion has its

main frequencies around 1 Hz [Schröder, 2023]. This is one order of magnitude slower than the acquisition of a single data point. However, there might still be some movement within 100 ms. This can be regarded as an increase of the actual measurement volume.

Third, there is the platform motion during the whole measurement run, which might last up to many hours. This leads to a motion of the focus, i. e. the point of measurement. This motion can be tracked using inertial measurement units (IMUs). For this reason, two IMUs in each telescope head are integrated in the measurement device. Whether this also allows for the correction of the tracked movement depends on the parameter of interest in the post-processing. For example, the mean wind velocity could be corrected for the platform motion. For other parameters it can be more intricate or even impossible. However, this is an error source that influences the analysis of the measured data but hardly the individual measurement data points. Therefore, a detailed analysis of the consequences of this platform motion is beyond the scope of this paper.

**Review:** *Therefore, as noted, this manuscript is somewhat limited to simply the propagation of the detector uncertainty to the measured wind in the lidar plane of reference, with an analysis of the filter/smoothing functions best suited to reduce this added noise. In this context, careful characterisation of the detector uncertainty would be important for the analysis. However, this characterisation is limited to just two sentences, which does not sufficiently justify the stated detector uncertainty of $\sigma^{det} = 0.04 \mathrm{ms}^{-1}$. From what I can tell, the authors take the resolution of the sensor output of $0.01 \mathrm{ms}^{-1}$ and assume a 99 % confidence bound(?) of $3\sigma$ to get a standard deviation of the uncertainty of $0.04 \mathrm{ms}^{-1}$ (assuming that they are rounding up?). However, this is just guesswork on my part. If this is the case, than the error propagation analysis is not even assessing the uncertainty of the individual lidar measurements, but is simply assessing the propagation of the resolution limitations of the individual lidar measurement.*

*I would therefore recommend that the authors, at the very least, provide more detail and care into the assessment and description of $\sigma^{det}$. The paper would also be much more strengthened by including additional error sources into their analysis, however this may require significant revisions of the manuscript.*

**Reply:** The main topic of the paper is, indeed, how each individual measurement error from three line-of-sight LiDAR sensors contributes to the measurement uncertainty of the three spatial components of the measured wind vector. Therefor, the individual measurement error $\sigma^{det}$ is the relevant quantity.

While our methodical approach is general and thus applicable for different values of $\sigma^{det}$, it should still be clear how we derive $\sigma^{det} = 0.04 \mathrm{\ ms}^{-1}$. We have added a new section 2.2 on the estimation of $\sigma^{det}$.

Also, it seems the reviewer accidentally used a sensor resolution of $0.01 \mathrm{\ ms}^{-1}$ instead of $0.1 \mathrm{\ ms}^{-1}$. May be this also contributed to the confusion?

**Review:** *Additional comments:*

1. **Review:** *Figure 2 was a little confusing for me due to the perspective. Specifically it took me some time to understand that the plane of the lidar was parallel to the oncoming wind field. I think the confusion comes from the kite being angled to the mean wind, but the lidar appearing to be drawn on the kite. Once I had figured out the arrangement of the CTL on the MPCK, the text of section 2.2 made more sense, but perhaps the authors may wish to add more details/different views to Figure 2 so that others may not be equally confused.*

   **Reply:** We thank the reviewer for this hint. We adapted Figure 2 accordingly, so that it is more intuitively clear now. Specifically, we have simplified the drawing so that only the lower part of the keel is shown, which now is perfectly aligned with the $x$-$z$-plane, as in reality.

2. **Review:** *Tables 1 and 2 have redundant information and could be combined. Note that whereas Table 2 refers to 10 Hz as the sampling rate, Table 1 refers to the same quantity as the time resolution. Technically, the time resolution is 0.1 s, not 10 Hz.*

   **Reply:** Thank you for the suggestion. We have considered combining the two tables. However Table 1 summarizes some basic specifications of the LiDAR setup whereas Table 2 specifies all modeling parameters. We therefore think that it is appropriate to keep the two separate. The only redundant information is indeed the sampling rate (we changed the wording to measurement rate as this is more appropriate). We have changed 'time resolution' to 'measurement rate' in Table 1 to make the presentation consistent.

3. **Review:** *line 182: should be 'lose', not 'loose'.*
   **Reply:** Thank you, we have corrected the typo.

4. **Review:** *4. The synthetic time series input the noise as a white noise process, what justification is there that the detector uncertainty appears in the form of white noise. Note that the nature of the noise could impact the efficacy of the smoothing for noise removal/uncertainty reduction.*
   **Reply:** Thank you for raising this question. We have added section 2.2 to the manuscript to describe the assumed measurement uncertainty in more detail.

   On the detector we have the coherent interference of the local oscillator with the signal. The local oscillator is adjusted so that it is the dominant noise term, hence we are in the shot noise limited detection scheme. Consequently, there are many photons arriving on the detector (more precisely: on each of the two balanced photo detectors). Also, having a peak coming out of the noise floor (otherwise the wind velocity can not be extracted, which can happen in reality, of course) means that there is also a significant number of signal photons involved. For large numbers of photons the Poisson distribution approximates the normal distribution very well.

   Now the subsequent question is: Does the dominance of white noise lead to an uncertainty of the peak position that is normally distributed? We have performed a simple simulation of a beat signal with added white noise, averaged 4000 spectra as in a typical measurement, and analysed the peak positions which were found by fitting a Gaussian function (see Appendix C). The result shows two things:

   (a) The peak positions and thus the velocities follow a normal distribution. It makes intuitively sense that the uncertainty of the peak position follows a symmetric function. The Poisson distribution is not symmetric for small mean values, but it approximates the normal distribution well for large mean values.

   (b) Increasing the level of added white noise, i. e. decreasing the signal-to-noise (SNR) ratio, leads to an increase in the fluctuation of the peak positions (the measured wind velocity). This shows that a decreasing SNR leads to increased uncertainty. However, it remains normally distributed for low SNR.

   Also, this is the main criticism mentioned by referee #1. As referee #1 had some additional comments on this matter, we would also like to refer to our reply to referee #1.

**References**

Marcel Schröder. *Cloud Microphysics Investigations with the Cloudkite Laboratory*. Phd thesis, Georg-August University Göttingen, Göttingen, March 2023. Available at https://hdl.handle.net/21.11116/0000-000D-06A7-0.